# PNeRV: A Polynomial Neural Representation for Videos

**Sonam Gupta**                                                                 *cs18d005@cse.iitm.ac.in*
*Department of Computer Science & Engineering, IIT Madras*

**Snehal Singh Tomar**                                                          *snehalstomar@gmail.com*
*Department of Electrical Engineering, IIT Madras*

**Grigorios G Chrysos**                                                         *chrysos@wisc.edu*
*University of Wisconsin-Madison*

**Sukhendu Das**                                                                *sdas@iitm.ac.in*
*Department of Computer Science & Engineering, IIT Madras*

**A. N. Rajagopalan**                                                           *raju@ee.iitm.ac.in*
*Department of Electrical Engineering, IIT Madras*

**Reviewed on OpenReview:** *https://openreview.net/forum?id=oCBsxCov2g*

## Abstract

Extracting Implicit Neural Representations (INRs) on video data poses unique challenges due to the additional temporal dimension. In the context of videos, INRs have predominantly relied on a frame-only parameterization, which sacrifices the spatiotemporal continuity observed in pixel-level (spatial) representations. To mitigate this, we introduce **P**olynomial **Ne**ural **R**epresentation for **V**ideos (PNeRV), a parameter-wise efficient, patch-wise INR for videos that preserves spatiotemporal continuity. PNeRV leverages the modeling capabilities of Polynomial Neural Networks to perform the modulation of a continuous spatial (patch) signal with a continuous time (frame) signal. We further propose a custom Hierarchical Patch-wise Spatial Sampling Scheme that ensures spatial continuity while retaining parameter efficiency. We also employ a carefully designed Positional Embedding methodology to further enhance PNeRV's performance. Our extensive experimentation demonstrates that PNeRV outperforms the baselines in conventional Implicit Neural Representation tasks like compression along with downstream applications that require spatiotemporal continuity in the underlying representation. PNeRV not only addresses the challenges posed by video data in the realm of INRs but also opens new avenues for advanced video processing and analysis.

## 1 Introduction

Implicit Neural Representations (INRs) have become the paradigm of choice for modelling discrete signals such as images and videos using a continuous and differentiable neural network, for instance, a multi layered perceptron. They facilitate several important applications like super-resolution, inpainting, and denoising (Niemeyer et al., 2019; Park et al., 2021; Pumarola et al., 2021; Tretschk et al., 2021; Xian et al., 2021; Li et al., 2021; Du et al., 2021) for images. They offer various important benefits over discrete representations particularly in terms of them being agnostic to resolution. Recent advancements have extended INR to video signals, but early methods relied on utilizing 3 dimensional spatiotemporal coordinates $(x, y, t)$ as input and RGB values as outputs. Such straightforward extensions of INRs to videos are inefficient during inference since they need to sample $T \times H \times W$ times to reconstruct the entire video. For high resolution videos, this behavior becomes more prominent.

Also, a simple multi layered perceptron is unable to model the complex spatio-temporal relationship in video pixels well. To address this issue and maintain parameter efficiency, current state-of-the-art methods in the field use a frame-only parameterization as depicted in Fig. 1 (a) and (b). These representations take the time index of a frame as input and predicts the entire frame as output. Although state-of-the-art INRs on video data exhibit impressive results on tasks such as video denoising and compression, they suffer from two fundamental issues. Firstly, the lack of spatial parameterization renders the representation less suitable for conventional INR applications such as video super-resolution. Secondly, they are not equipped to capture the information pertaining to pixel-wise auto and cross correlations across time explicitly. Hence, resulting in a suboptimal metric performance to model size ratio. Only recently, Sen et al. (2022) have attempted to explore a spatiotemporally continuous neural representation based hypernetwork for generating videos. However, their approach and the tasks they enable are fundamentally different[1] to ours.

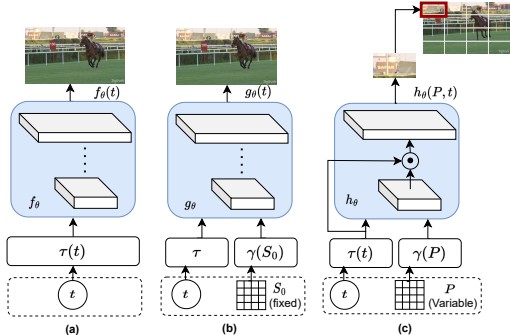

Figure 1: **PNeRV when compared to its counterparts:** (a) NeRV: An INR for videos with only frame-wise parameterization that leads to loss of spatial continuity. (b) E-NeRV: A step-up over NeRV with a parameterization that employs a fixed Spatial Context (SC). The fixed SC does not support spatial continuity. (c) **PNeRV:** An efficient INR for videos with a PNN backbone (signified by the usage of Hadamard Product $\odot$) that supports varying SC while retaining spatial continuity.

We utilize the following key insights to build a spatiotemporally continuous Neural Representation while keeping the model size in check: (1) Achieving spatiotemporal continuity doesn't always require dense per-pixel sampling. A well-designed patch-wise sampling approach (Tretschk et al., 2020; Yuval Nirkin, 2021) can yield comparable results for downstream tasks while processing less data. (2) To achieve better efficiency in handling higher-dimensional inputs with fewer learnable parameters and maintaining performance, we consider using Polynomial Neural Networks (PNNs) (Chrysos et al., 2021b; 2019) as our preferred function approximator. PNNs model the auto and cross correlations within their input feature maps. (3) We also propose a Positional Embedding (PE) methodology to aid the PNN backbone in learning a faithful representation using the sampled inputs. Carefully designed PEs (Vaswani et al., 2017; Wu et al., 2021; Deng et al., 2022; Sitzmann et al., 2020b) are proven to boost the performance of Deep Neural Networks.

In this work, we enhance INRs for videos along the following three directions. Firstly, we adopt a temporal as well as spatial parameterization (illustrated in Fig. 1(c)) in our light-weight representation. We achieve this by replacing the dense pixel-wise spatial sampling with a carefully designed Hierarchical Patch-wise Spatial Sampling approach. Our scheme (elaborated upon in section 3.1) breaks a video frame into patches and samples coordinates from sub-patches in a recursive fashion across different levels of hierarchy. Secondly, we leverage the properties of PNNs to build a parameter-wise efficient decoder backbone that yields better metric performance. PNeRV also inherits some important properties of PNNs such as robustness to the choice of non-linear activation functions. Finally, we improve the positional embedding of input signals to align well with our PNN backbone and achieve peak metric performance. Our claims are backed by consistent qualitative and quantitative results on video reconstruction and four challenging downstream tasks i.e. Video Compression, Super-Resolution, Frame Interpolation, and Denoising. The key contributions of this paper can be summarized as:

1. We introduce a Hierarchical Patch-wise Spatial Sampling approach in our formulation which makes PNeRV continuous in space and time while retaining parameter efficiency.

2. We design a PNN for temporal signals. We build a Higher order Multiplicative Fusion (HMF) module that learns parametric embedding.

---

[1]We highlight these differences in section 2.

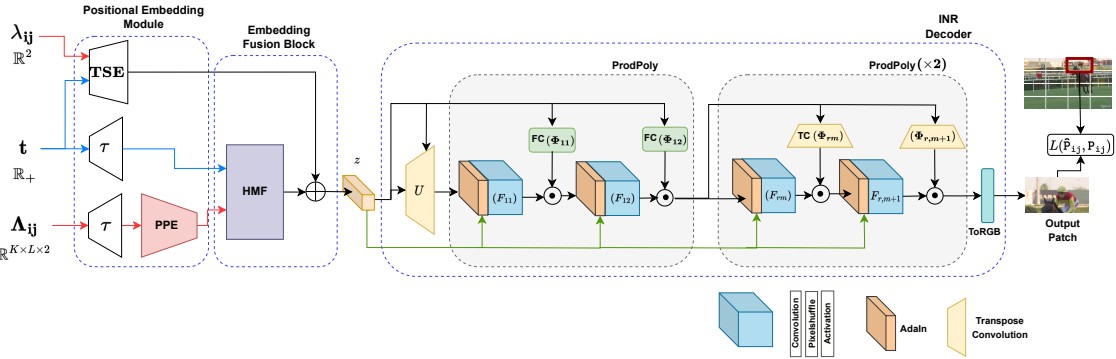

Figure 2: **The PNeRV Architecture**: The PNeRV pipeline consists of three modules. First, the PEs of time index $t$, coarse patch coordinate $\boldsymbol{\lambda_{ij}}$ and the fine patch coordinate $\boldsymbol{\Lambda_{ij}}$ are computed in the Positional Embedding Module. Second, these embeddings are fused effectively in the Embedding Fusion Block. Finally the PNN-based INR decoder reconstructs the frame patch, given a fused Positional Embedding $z$. Here FC denotes a fully connected layer of appropriate input-output dimensions.

3. We propose a new positional embedding scheme to encode and fuse spatial and temporal signals. The scheme brings together both parametric (learnable) and functional (deterministic) embeddings, a first in Neural Representations for videos. We show that both the embeddings complement each other to align well with the PNN based backbone and attain peak metric performance.

## 2 Related Work

**Implicit Neural Representations.** INR is a method to convert conventionally discrete signal representations such as images (discrete in space) and videos (discrete in space and time) into continuous representations. Originally motivated as an alternate representation for images (Park et al., 2019; Mescheder et al., 2019; Chen & Zhang, 2019), INR has been pushing the envelope in terms of performance on a wide array of tasks on images such as denoising and compression (Zhu et al., 2022; Huang et al., 2022; Li et al., 2022a; Chen et al., 2022a). INR for videos extends INR for images by a simple reparameterization in terms of video-frame indices as well (Niemeyer et al., 2019; Park et al., 2021; Pumarola et al., 2021; Tretschk et al., 2021; Xian et al., 2021; Li et al., 2021; Du et al., 2021; Chen et al., 2022b; Saragadam et al., 2022; Mai & Liu, 2022). The approach of choice for such architectures entails learning an embedding for pixels and timestamps, which are passed on to a decoder network. To expedite model training and inference with large video tensors in such INR formulations, state-of-the-art literature in INR for videos (Chen et al., 2021; Li et al., 2022b; Chen et al., 2023) has introduced parameterization over frame indices only. While such formulations are lighter and faster, they compromise spatial continuity. We aim to bring the best of both these formulations together in this work by employing a parameterization over patches as well as frame indices, with a PNN backbone. Consequentially, the spatial continuity achieved while keeping model parameters in check, is an essential attribute for a faithful INR and is critical for applications such as super-resolution.

(Sen et al., 2022) have recently attempted to build a spatiotemporally continuous INR based hypernetwork for generating videos. Their proposed method differs from ours in two key aspects. First, theirs is a *video generation* pipeline and the INR is only a *component* of their model. Whereas ours is a vanilla INR that serves as an alternate representation for videos while enabling interesting downstream tasks. Second, since their model is a hypernetwork, it is not well equipped to tackle high resolution videos such as the ones found in the UVG dataset (Mercat et al., 2020). The authors attribute this behaviour to the unstable training routines of large hypernetworks.

**Polynomial Neural Networks (PNNs).** PNNs model their outputs as a higher-degree polynomial of the input. A full polynomial expansion can be expressed as follows (Chrysos et al., 2021b):

Table 1: Overview of the nomenclature used in section 3. All PEs $\in \mathbb{R}^{1 \times 2\mathbf{l}}$, where $\mathbf{l}$ is a hyperparameter.

| Nomenclature Pertaining to Spatial Sampling (Section 3.1) | | | Nomenclature Pertaining to PEs (Section 3.2) | |
|---|---|---|---|---|
| Symbol | Dimension | Definition | Symbol | Definition |
| $V$ | $T \times H \times D \times 3$ | Complete Video | $b$ | Hyperparameter: frequency. |
| $v_t$ | $H \times D \times 3$ | $t^{th}$ frame | $\mathbf{l}$ | Hyperparameter: length of the PE. |
| $\mathcal{C}$ | $H \times D \times 2$ | Global spatial (pixel) coordinates. | $\mathbf{\Gamma}_{TSE}(\boldsymbol{\lambda}_{ij}, t)$ | Functional embedding to fuse space and time. |
| $\boldsymbol{\lambda}_{ij}$ | 2 | Grid coordinate in $\mathcal{C}$ corresponding to $\boldsymbol{P}_{ij}$. | $\mathbf{\Gamma}_{HMF}(\boldsymbol{\Lambda}_{ij}, t)$ | PNN driven fusion of $\mathbf{\Gamma}_{FPE}(t)$ and $\mathbf{\Gamma}_{PPE}(\boldsymbol{\Lambda}_{ij})$ |
| $\boldsymbol{\Lambda}_{ij}$ | $K \times L \times 2$ | Tensor with fine coordinates in $\mathcal{C}$ that correspond to $\tilde{\boldsymbol{P}}_{kl}$. | $\mathbf{\Gamma}_{ijt}$ | INR Decoder input: $(\mathbf{\Gamma}_{TSE}(\boldsymbol{\lambda}_{ij}, t) + \mathbf{\Gamma}_{HMF}(\boldsymbol{\Lambda}_{ij}, t))$ |
| $\boldsymbol{P}_{ij}$ | $\frac{H}{M} \times \frac{D}{N} \times 3$ | $(i,j)^{\text{th}}$ Coarse patch. $M \times N$ such patches are sampled $\forall v_t$. | $\mathbf{\Gamma}_{FPE}(t)$ | Functional PE of $t$. |
| $\tilde{\boldsymbol{P}}_{kl}$ | $\frac{H}{MK} \times \frac{D}{NL} \times 3$ | $(k,l)^{\text{th}}$ fine patch. $K \times L$ such fine patches are sampled $\forall \boldsymbol{P}_{ij}$. | $\mathbf{\Gamma}_{PPE}(\boldsymbol{\Lambda}_{ij})$ | Parametric embedding of $\boldsymbol{\Lambda}_{ij}$ |

$$\boldsymbol{x} = \sigma(\boldsymbol{W_1^T} \boldsymbol{z} + \boldsymbol{z}^T \boldsymbol{W_2} \boldsymbol{z} + \boldsymbol{\mathcal{W}_3} \times_1 \boldsymbol{z} \times_2 \boldsymbol{z} \times_3 \boldsymbol{z} + ... + \boldsymbol{b}), \tag{1}$$

where, $\boldsymbol{x}$, $\boldsymbol{z}$, $\sigma$, and $\boldsymbol{b}$ represent the output, input vector, non-linear activation and bias. $\boldsymbol{\mathcal{W}_i}$ represents the weight tensor for the $i^{\text{th}}$ order, and $\times_i$ represents the *mode-i* product[2]. The PNN paradigm's elegance lies in the utilization of tensor factorization techniques to prevent an exponential increase in model parameters with an increase in the polynomial order. We examine only the Nested Coupled CP Decomposition (NCP)[3] since our model implementation is based on its sequential polynomial expansion. Considering a $3^{\text{rd}}$ order polynomial governed by Eq. 1, the decomposed forward pass can be expressed as the following recursive relationship:

$$\boldsymbol{x}_n = (\boldsymbol{A}_{[n]}^T \boldsymbol{z}) \odot (\boldsymbol{S}_{[n]}^T \boldsymbol{x}_{n-1} + \boldsymbol{B}_n^T \boldsymbol{b}_{[n]}), \tag{2}$$

for $n \in \{2, 3\}$. Here $\boldsymbol{x} = \boldsymbol{C}\boldsymbol{x}_3 + \boldsymbol{q}$ is the output of the $3^{\text{rd}}$ order polynomial, $\odot$ represents Hadamard product and $\boldsymbol{x}_1 = (\boldsymbol{A}_{[1]}^T \boldsymbol{z}) \odot (\boldsymbol{B}_1^T \boldsymbol{b}_{[1]})$. The learnable parameters in this setup are $\boldsymbol{C} \in \mathbb{R}^{o \times k}, \boldsymbol{A}_{[n]} \in \mathbb{R}^{d \times k}$, $\boldsymbol{S}_{[n]} \in \mathbb{R}^{k \times k}$, $\boldsymbol{B}_{[n]} \in \mathbb{R}^{e \times k}$, and $\boldsymbol{b}_{[n]} \in \mathbb{R}^e$, and $\boldsymbol{q} \in \mathbb{R}^o$. The symbols $d$, $o$, $e$, and $k$ represent the decomposition's input dimensions, output dimensions, implicit dimension, and rank. The rise of PNNs has seen their application to an array of important deep learning regimes such as generative models (Chrysos et al., 2021b; Choraria et al., 2022; Singh et al., 2023), attention mechanisms (Babiloni et al., 2021), and classification models (Chrysos et al., 2022a;b; 2023; Chen et al., 2024). However, their direct application to temporal signals has not emerged, and they have only been used in a single variable setup in unconditional modeling regimes. PNeRV builds along these new directions in its INR decoder and HMF.

**Rich Positional Embeddings.** PEs based on a series of sinusoidal functions much like the Fourier series, have become an integral part of INRs. Several works (Mildenhall et al., 2021; Sitzmann et al., 2020a; Tancik et al., 2020) have shown that in the absence of such embeddings, the output of the INR is blurry i.e. misses the high frequency information. Thus, PEs enable INRs to capture fine-details of a signal making them indispensable for image applications (Wu et al., 2021; Deng et al., 2022; Skorokhodov et al., 2021). INR methods for videos have also sought to capitalize upon the advantages of an efficient PE (Sitzmann et al., 2020b; Mai & Liu, 2022). However, state-of-the-art in the domain (Li et al., 2022b; Chen et al., 2021) has only explored functional (deterministic) embeddings in one input variable. In contrast, PNeRV employs both parametric (learnable) and functional embeddings. We also introduce a PNN based fusion strategy to combine the functional and parametric embeddings.

## 3 PNeRV: Polynomial Neural Representation for Videos

**Overview:** Let us now introduce our method. The notation and definitions for the various elements used in this section is summarized in Table 1. We denote tensors by calligraphic letters, matrices by uppercase boldface letters and vectors by lowercase boldface letters. To enable spatial continuity while keeping the model size in check, we propose a Hierarchical Patch-wise Spatial Sampling approach for the input coordinates.

As shown in Fig. 2, the PNeRV architecture comprises three key components, namely, a Positional Embedding Module, an Embedding Fusion Block, and the PNN-based INR decoder. Each frame $v_t$ in an input

---

[2]Defined in appendix A.2.
[3]Definition adopted from Chrysos et al. (2021b).

video $V = \{v_t\}_{t=1}^T$ is recursively divided into coarse patches and fine sub-patches. Coordinates sampled from both the patch and sub-patch instances along with their respective frame index ($t$) serve as inputs to the INR decoder. In nutshell, the PNeRV formulation can be represented as:

$$P_{ij} = F_{\Theta}(\Lambda_{ij}, \lambda_{ij}, t), \tag{3}$$

where, $F_{\Theta}$ denotes the complete PNeRV model (having parameters $\Theta$). As defined in Table 1, $\Lambda_{ij}$ denotes a fine coordinate Tensor, $\lambda_{ij}$ is a coarse patch coordinate, and $t$ is the frame index. We present a detailed discussion on each of our model's constituent elements in the subsections that follow.

## 3.1 Hierarchical Patch-wise Spatial Sampling

State-of-the-art methods Chen et al. (2023; 2021); Li et al. (2021) have drifted away from a spatial parameterization of their representation to ensure faster inference. They resort to a temporal-only parameterization. In contrast, PNeRV uses a spatiotemporal parameterization whilst having fewer parameters by employing our efficient sampling approach (depicted in Fig. 3). We observed that a pixelwise formulation increases the computational complexity manifold. Hence, we opt for a hierarchical patch-wise formulation. A primitive method to sample spatial patch coordinates would be to assign a scalar coordinate to each patch (similar to frame indices). However, the pitfalls of such an approach are twofold. Firstly, scalar patch indices lack spatial context. They do not convey any sense of spatial localization. Secondly, PEs obtained from scalars have a lower variance, which is not ideal for training. Our

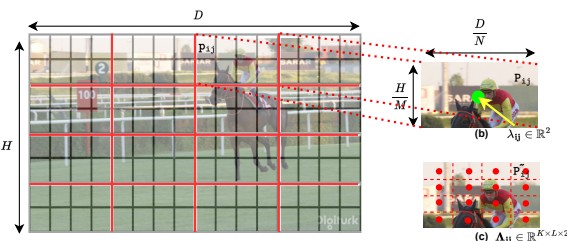

Figure 3: **Hierarchical Patch-wise Spatial Sampling:** (a) A Global coordinate grid $\mathcal{C}$ with input values normalized to range $[0, 1]$ is constructed for each frame. (b) The grid is divided into $M \times N$ coarse patches of equal size. For a coarse patch $P_{ij}$, its centroid is used as a 2D coordinate $\lambda_{ij}$. (c) Each coarse patch is further divided into $K \times L$ fine patches and a collection of the centroids of these smaller patches is used as the fine patch coordinate tensor $\Lambda_{ij}$.

analysis in Table 8 underscores these pitfalls. We have designed our sampling strategy to enrich the input to our INR decoder with spatial information of the patches. Instead of associating just a scalar index to each patch, we associate each patch $P_{ij}$ with a coarse 2D index $\lambda_{ij}$ and a fine index $\Lambda_{ij} \in \mathbb{R}^{K \times L \times 2}$. The process of computing $\lambda_{ij}$ and $\Lambda_{ij}$ is illustrated in Fig. 3. Like traditional INRs Niemeyer et al. (2019); Park et al. (2021); Pumarola et al. (2021), we first build a global coordinate grid $\mathcal{C}$ of size $H \times D$ normalized to range $[0, 1]$ (Fig. 3 (a)). Next, each frame is divided into $M \times N$ coarse patches. The coordinates $\lambda_{ij}$ for these coarse patches $P_{ij}$ are found by computing their centroids (Fig. 3 (b)). Further, each coarse $P_{ij}$ is divided into $K \times L$ fine sub-patches. The $K \times L \times 2$ dimensional tensor formed by the centroids of each of these sub-patches is used as the fine coordinates of $P_{ij}$ (Fig. 3 (c)). It is imperative to note that, although we divide a frame into patches, the normalized coordinate values are sampled from $\mathcal{C}$ in all cases for computation of centroids. In effect, the manner in which the patch-coordinates are sampled in our scheme is hierarchical in nature. This ensures a sense of spatial locality in all patches. Intuitively, the coarse coordinate captures a global context whilst the fine coordinates of a patch capture the local context. Algorithm 1 in Appendix A.3 summarizes hierarchical patch-wise spatial sampling.

## 3.2 Positional Embedding Module

Literature on INRs (Sitzmann et al., 2020b; Tancik et al., 2020) dictates that rich positional embeddings (PEs) are central to the performance of INR methods. Fourier series like PEs are positively correlated with the network's ability to capture the high frequency information Tancik et al. (2020). Although the field has witnessed several advances toward the development of optimal functional (fixed) embeddings of signals and their parametric (learnable) fusion, functional fusion and parametric embeddings remain under explored. In this work, we exploit the combination of functional PEs, parametric PEs, functional PE fusion, and parametric PE fusion to learn a superior INR for videos.

We propose an embedding scheme wherein we perform a temporal functional embedding in $t$, a spatial embedding via functional fusion, and a parametric (multiplicative) fusion of all PEs to yield a rich spatiotemporally aware PE. We elaborate upon each of our embeddings and their parametric fusion in the sections that follow.

**Positional Encoding of Frame Index (FPE)** Given a frame index $t$, normalized between $[0, 1]$ as input, we adopt the widely used Fourier series based positional encoding scheme similar to the existing methods Chen et al. (2021); Li et al. (2022b). This embedding is given as:

$$\mathbf{\Gamma}_{FPE}(t) = [\sin(\pi\nu^i t) \quad \cos(\pi\nu^i t) ...]_{i=0}^{1-1}, \quad (4)$$

where, $\nu$ denotes the frequency governing hyperparameter and $\mathtt{l}$ governs the number of sinusoids.

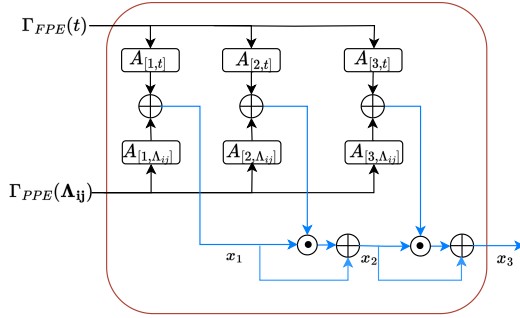

Figure 4: **The HMF architecture at a glance:** All linear transformation matrices represent the terms in Eq. 9. Here, $\odot$ denotes the Hadamard Product, $\oplus$ represents feature addition, black arrows represent inputs, and blue arrows represent the fused entities.

**Parametric Embedding of Fine Coordinates (PPE)** We employ a parametric positional embedding scheme (PPE) to encode the spatial context available in the fine patch coordinates given by tensors $\mathbf{\Lambda}_{ij}$. The PPE block in Fig. 2 illustrates the same. First, Eq. 4 is applied to each element of $\mathbf{\Lambda}_{ij}$ to map it to $\mathbb{R}^{1 \times 21}$ dimensional vectors. These resultant embeddings are arranged side by side in spatial order to obtain a feature map of size $\mathbb{R}^{K \times L \times 41}$. Notice that each value in the $K \times L$ grid has a 2D coordinate value corresponding to $x$ and $y$. Eq. 4 is applied individually to the $x$ and $y$ coordinates and the resulting vectors are fused across the channel dimensions. Resulting in a channel dimension of $4l$. To merge these features we use a Non-Local Block Wang et al. (2018) followed by a linear layer. This spatially aware attention based fusion mechanism encourages a weighted feature fusion between various spatial regions where the weights are governed by the Non-Local Block. We refer to this parameterized embedding as $\mathbf{\Gamma}_{PPE}(\mathbf{\Lambda}_{ij})$.

**Time Aware Spatial Embedding (TSE)** A video can be seen as time modulated spatial signal. Therefore, ideally, the spatial positional embedding should be dependent on the frame-index (time) as well as patch coordinates. To this end, we design a Time Aware Spatial Embedding which is inspired from Angle modulation. In analog communication, Angle Modulation refers to the technique of varying a carrier signal's phase in accordance with the information content of a modulating signal. The general expression for the same is given by

$$y_c(t) = Amp_c\{\cos(2\pi f_c t) + \phi(\cos(2\pi f_m t))\}, \quad (5)$$

where, $y_c$ is the modulated signal, $Amp_c$ is the amplitude of the carrier signal, $\phi(.)$ is the phase governing function. $f_c$ and $f_m$ are the frequencies of the carrier and modulated signals, respectively. We design the embedding to perform functional fusion of $\boldsymbol{\lambda}_{ij}$ and $t$. We model a video as a time ($t$) modulated spatial signal ($\boldsymbol{\lambda}_{ij}$). The proposed embedding (denoted by $\mathbf{\Gamma}_{TSE}$) is governed by Eqs. 6 and 7.

$$\mathbf{\Gamma}_{TSE}(\boldsymbol{\lambda}_{ij}, t) = [\cos(\Omega_{ij}^\alpha t) \quad \sin(\Omega_{ij}^\alpha t) ...]_{\alpha=0}^{1-1}, \quad (6)$$

wherein,

$$\Omega_{ij}^\alpha = 2\pi\beta^\alpha + \frac{\sin(2\pi\lambda_{xij}\beta^\alpha)}{\beta^\alpha} + \frac{\sin(2\pi\lambda_{yij}\beta^\alpha)}{\beta^\alpha}. \quad (7)$$

Our ablations (Table 8) substantiate that functional fusion ($\mathbf{\Gamma}_{TSE}$) complements parametric fusion of $\mathbf{\Gamma}_{FPE}(t)$ and $\mathbf{\Gamma}_{PPE}(\mathbf{\Lambda}_{ij})$ to boost performance.

### 3.3 Embedding Fusion Block

Effective fusion of all our positional embedding elements is critical to the performance of our method. We opt for a hybrid functional and parametric fusion module to bring together the positional embeddings obtained

via the $\mathbf{\Gamma}_{FPE}(.), \mathbf{\Gamma}_{PPE}(.)$, and $\mathbf{\Gamma}_{TSE}(.)$ functions. Our fusion mechanism is split over two stages. First $\mathbf{\Gamma}_{FPE}(t)$ and $\mathbf{\Gamma}_{PPE}(\mathbf{\Lambda}_{ij})$ are fused using our proposed Higher-order Multiplicative Fusion (HMF) block. Then, $\mathbf{\Gamma}_{TSE}(\boldsymbol{\lambda}_{ij}, t)$ is added to the resulting vector, resulting in new embedding $\boldsymbol{z}$ that acts as input to the INR decoder.

**Higher-order Multiplicative Fusion (HMF)** We introduce the HMF which is a Nested-CoPE (Chrysos et al., 2021a) inspired fusion mechanism, to fuse $\mathbf{\Gamma}_{FPE}(t)$ and $\mathbf{\Gamma}_{PPE}(\mathbf{\Lambda}_{ij})$. As shown in Fig. 4, HMF entails additive fusion of the linearly transformed fusion entities to capture first-order correlations. The additive fusion blocks are followed by a Hadamard product operation with the previous additive fusion output in a recursive fashion for three iterations. The recursive structure ensures that cross-correlations are captured well by the fused output. The fusion in effect translates to the following recursive relationship:

$$\boldsymbol{x}_n = ((\boldsymbol{A}_{[n,t]}^T \mathbf{\Gamma}_{FPE}(t) + \boldsymbol{A}_{[n,\mathbf{\Lambda}_{ij}]}^T \mathbf{\Gamma}_{PPE}(\mathbf{\Lambda}_{ij})) \odot \boldsymbol{x}_{n-1}) + \boldsymbol{x}_{n-1}, \tag{8}$$

wherein,

$$\boldsymbol{x}_1 = \boldsymbol{A}_{[1,\mathbf{\Lambda}_{ij}]}^T \mathbf{\Gamma}_{PPE}(\mathbf{\Lambda}_{ij}) + \boldsymbol{A}_{[1,t]}^T \mathbf{\Gamma}_{FPE}(t).$$

Here, $n \in \{2, 3\}$, $\boldsymbol{x}_3$ represents the fused embedding (output of HMF block), and $\odot$ represents Hadamard product. The learnable parameters in HMF are $\boldsymbol{A}_{[n,T]} \in \mathbb{R}^{2l \times k}$ and $\boldsymbol{A}_{[n,\mathbf{\Lambda}_{ij}]} \in \mathbb{R}^{2l \times k}$. The rank of the decomposed weight matrices $k$, is taken to be 160. As highlighted in Chrysos et al. (2021a), the adopted approach for fusing the frame-timestamps and patches has an advantage over a standard approach that employs concatenation followed by downsampling. In that, concatenation amounts to the additive format of fusion which fails to capture cross-terms in correlation. That is, multiplicative interactions of order 2 or more are essential for capturing both auto and cross-correlations among the entities to be fused.

### 3.4 INR Decoder

The literature on PNNs Chrysos et al. (2021b) has shown that stacking two or more polynomials in a multiplicative fashion leads to a desired order of the underlying polynomial with much lesser parameters. Such an approach is termed as *ProdPoly* (Product of Polynomials). As defined by (Chrysos et al., 2021b), a ProdPoly implementation entails the Hadamard product of outputs of sub-modules in the architecture to obtain a higher order polynomial in the input. Since the order of a polynomial is directly correlated with its modelling capabilities, the ProdPoly approach is suitable for designing our lightweight INR decoder. The proposed INR decoder is a modified derivative of the ProdPoly formulation. In that, we design the INR decoder as a product of three polynomials. Per our formulation, the output of the $r^{\text{th}}$ polynomial is given as input to the $(r + 1)^{\text{th}}$ block. The advantage of such a stacking is that it leads to an exponential increase in order of the polynomial.

Specifically, we have three ProdPoly blocks in a hierarchy. The first ProdPoly block accepts as the fused embedding $\boldsymbol{z}$ as input. The other two ProdPoly blocks take the output feature map from their preceding ProdPoly block, $\boldsymbol{o}_{r-1}$ as their input (Fig. 2). Each ProdPoly block in INR decoder is an adapted implementation of an NCP decomposed PNN variant tailored to our model's requirement. The NCP-polynomial in each ProdPoly block is implemented using two convolutional blocks $F$. The design of these blocks is inspired by Chen et al. (2021); Li et al. (2022b). Each $F$ block entails an Adaptive Instance Normalization layer (AdaIn) Karras et al. (2019), Convolution, pixel shuffle operation and a GeLU Hendrycks & Gimpel (2016) activation layer. This operation is denoted as $F(.)$. The AdaIn layer takes $\boldsymbol{z}$ as input and normalizes the feature distribution with spatio-temporal context embedded in the input vector $\boldsymbol{z}$. In essence, we adapt Eq. 2 the following, for our decoder where $S$ and $A$ are implemented as $F$ and $\mathbf{\Phi}$ :

$$\boldsymbol{y}_{rm} = F_{rm}(\boldsymbol{y}_{rm-1}) \odot (\mathbf{\Psi}_{[rm]}^T \boldsymbol{r}_i) \, ; \, m \in \{1, 2\}, \tag{9}$$

wherein,

$$\boldsymbol{y}_{r1} = (F_{r1}(\boldsymbol{U}^T \boldsymbol{o}_r)) \odot (\mathbf{\Psi}_{[r1]}^T \boldsymbol{z}),$$

$\boldsymbol{o}_r = \boldsymbol{y}_{r2}$ is the output of $r^{\text{th}}$ ProdPoly block. $\boldsymbol{U}$ is a set of three transpose convolutional layers applied only before the first ProdPoly block to obtain a 2D feature map from the input vector $\boldsymbol{z}$. $\boldsymbol{o}_3$ is the final output

Table 2: **Quantitative comparisons** in terms of PSNR (dB) with respect to reconstruction on the Scikit-Bunny video and the UVG datset. PNeRV achieves state-of-the-art performance while maintaining significantly fewer parameters and being up to $4\times$ faster in terms of rate of convergence.

| Method | # Params (M)↓ | Bunny | Beauty | Bosphorus | Bee | Jockey | SetGo | Shake | Yacht |
|---|---|---|---|---|---|---|---|---|---|
| NeRV-L | 12.57 | 39.63 | 36.06 | 37.35 | 41.23 | 38.14 | 31.86 | 37.22 | 32.45 |
| HNeRV | 11.90 | 36.23 | 36.17 | 30.20 | 41.58 | 28.55 | 29.67 | 32.44 | 25.50 |
| E-NeRV | 12.49 | 42.87 | 36.72 | 40.06 | 41.74 | 39.35 | 34.68 | 39.32 | 35.58 |
| **Ours** | **11.89** | **44.90** | **39.8** | **41.86** | **43.98** | **39.84** | **35.82** | **41.37** | **36.93** |
| **Gain over E-NeRV** | ↓ **0.6** | ↑ **2.03** | ↑ **3.08** | ↑ **1.8** | ↑ **2.24** | ↑ **0.49** | ↑ **1.14** | ↑ **2.05** | ↑ **1.35** |

(i.e. reconstructed patch $\hat{\mathbf{p}}_{ij}$) of the INR decoder. $\mathbf{\Psi}_{1m}$'s in the first ProdPoly block are implemented as linear layers. In the remaining blocks, transpose convolution layer is used with appropriate padding and strides. To remove the redundant parameters, similar to Li et al. (2022b), we also replace the convolutional kernel in $F_1$ with two consecutive convolution kernels with small channels. The optimal rank for our resultant polynomial's decomposition per the NCP (Eq. 2) was found to be 324. Appendix A.4 presents a detailed study pertaining to the choice of optimal rank for the decomposition, alongside elaborate architecture details.

### 3.5 Training

To train our network, we randomly sample a batch of frame patches $\boldsymbol{P}_{ij}$ along with their normalized fine coordinates, coarse coordinates, and the time indices $(\boldsymbol{\Lambda}_{ij}, \boldsymbol{\lambda}_{ij}, t)$. These indices are then given as input to PNeRV to predict the corresponding patches $\hat{\boldsymbol{P}}_{ij}$. The model is trained by using a combination of the L1 and SSIM Wang et al. (2004) losses between the predicted frame patches and ground truth frame patches, governed by Eq. 10

$$L(\hat{\boldsymbol{P}}_{ij}, \boldsymbol{P}_{ij}) = \frac{1}{M \times N \times T} \sum_{t=1}^{T} \sum_{p=1}^{M \times N} \gamma ||\hat{\boldsymbol{P}}_{ij} - \boldsymbol{P}_{ij}||_1 + (1 - \gamma)(1 - SSIM(\hat{\boldsymbol{P}}_{ij}, \boldsymbol{P}_{ij})) \tag{10}$$

where, $M \times N$ is the total number of patches per frame, $T$ denotes the total number of frames, and $\gamma$ is a hyper-parameter to weigh the loss components. We set $\gamma$ to 0.7. We infer frame patches at all the locations and concatenate them in a consistent manner to reconstruct the original videos. Since the model learns non-overlapping patches independently, the intensity changes near the patch edges may cause the reconstructed frames to have boundary artifacts. We apply Gaussian blur to the reconstructed video to mitigate these subtle artifacts. No further post-processing is required for continuity and coherence in the generated frames.

## 4 Experiments

We split our experimental analysis of PNeRV into (1) evaluation of the representation ability using Video Reconstruction task (2) testing the efficacy on the proposed downstream tasks (3) performing appropriate ablation studies to assess the contributions and salience of individual design elements. The downstream tasks we perform include (i) *Video Compression* to assess the applicability of PNeRV as an alternate lightweight video representation (ii) *Video Super-Resolution* to assess the spatial continuity of PNeRV (iii) *Video Interpolation* to assess the temporal continuity of PNeRV (iv) *Video Denoising* as an interesting application of PNeRV. We also compare the rate of convergence (during training) of PNeRV vis-à-vis prior art.

**Experimental Setup:** We train and evaluate our model on the widely used UVG dataset (Mercat et al., 2020) and the "Big Buck Bunny" (Bunny) video sequence from scikit-video. The UVG dataset comprises 7 videos. Each UVG video is resized to $720 \times 1280$ resolution and every $4^{th}$ frame is sampled such that the entire video contains 150 frames. All 132 frames of the Bunny sequence are used at a resolution of $720 \times 1280$. For all our experiments, we train each model for 300 epochs with a batch size 16 (unless specified otherwise) with up-scale factors set to $5, 2, 2$. The input embeddings $\mathbf{\Gamma}_{FPE}$, $\mathbf{\Gamma}_{TSE}$, and $\mathbf{\Gamma}_{PPE}$ are computed with $\nu = 1.25$. We set $l = 80$ for $\mathbf{\Gamma}_{FPE}$ and $\mathbf{\Gamma}_{TSE}$. Whereas, $\mathbf{\Gamma}_{TSE}$ uses $\alpha = 40$. The network is trained using

Adam optimizer (Kingma & Ba, 2014) with default hyperparameters, a learning rate of $5e^{-4}$, and a cosine annealing learning rate scheduler (Loshchilov & Hutter, 2016). Following E-NeRV's evaluation methodology, we use PSNR (Wang et al., 2003) to evaluate the quality of the reconstructed videos.

## 4.1 Video Reconstruction

High fidelity video reconstruction assumes utmost importance when it comes to building an INR. We compare PNeRV with several state-of-the-art methods, namely NeRV-L (Chen et al., 2021), E-NeRV (Li et al., 2022b) and HNeRV (Chen et al., 2023) on videos belonging to the UVG dataset and the Bunny video. The PSNR values obtained for reconstructed videos are reported in Table 2. We observe that our model consistently outperforms existing methods on a diverse set of videos, while employing significantly lesser number of learnable parameters shows improvements on videos with slow moving objects like Beauty, Bee, Shake as well as dynamic videos like Bunny, Bosphorus and Yacht. Hence, validating that the PNN-backed PNeRV is a lightweight INR that captures the necessary spatiotemporal correlations needed to better represent videos. We present qualitative comparisons with state-of-the-art for the task in Fig. 10 (Appendix A.6) (left column). Appendix A.5 presents additional qualitative results.

## 4.2 Downstream Tasks

### 4.2.1 Video Compression

Recent video compression algorithms follow a hybrid approach where a part of the compression pipeline consists of neural networks while following the traditional compression pipeline (Agustsson et al., 2020; Yang et al., 2020; Wu et al., 2018). An INR encodes a video as the weights of a neural network. This enables the use of standard model compression techniques for video compression. Following (Chen et al., 2021), we employ model pruning for video compression. We present experimental results for the same on the "Big Buck Bunny" sequence from scikit-video in Figure 5. It can be observed that a PNeRV model of 40% sparsity achieves results comparable to the full model, in terms of reconstruction accuracy and perceptual coherence. Fig. 10 (Appendix A.6) (middle column) presents qualitative comparisons with state-of-the-art for the task. For sparsity values less than 45%, our model outperforms NeRV and E-NeRV. However, beyond 45% sparsity, PNeRV's performance de-

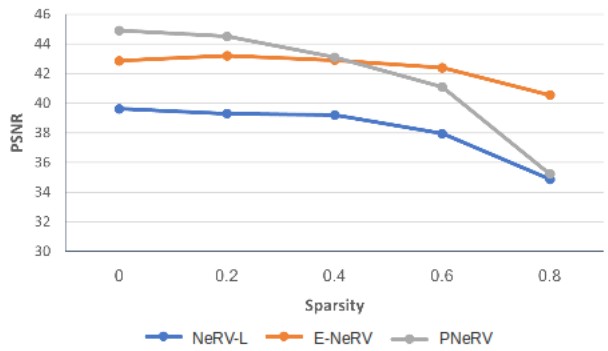

Figure 5: Model pruning results on NeRV-L, E-NeRV and PNeRV trained for 300 epochs on "Big Buck Bunny" video. Sparsity represents the ratio of pruned parameters.

grades rapidly. This behaviour can be attributed to the use of multiplicative interactions in PNeRV which cause model performance to increase rapidly with increase in model parameters. We provide additional qualitative results, quantitative results on the UVG dataset, and comparisons with HNeRV in Appendix A.7. From Fig. 17, it can be observed that the frames predicted by HNeRV are blurred , a typical property of autoencoder type of an architecture whereas our method is able to preserve the fine details well.

### 4.2.2 Video Super-Resolution

We present qualitative results for $\times 4$ Super-Resolution in Fig. 6. As reported in Table 3, for Super-Resolution, we compare our results with bicubic interpolation, ZSSR (Assaf Shocher, 2018), and SIREN (Sitzmann et al., 2020b). PNeRV outperforms these baselines in each case, which confirms that PNeRV is a generic spatiotemporal representation that lends itself well to various downstream tasks that require spatial continuity without the need for task-specific retraining or fine-tuning. We also provide reasons for not comparing our results with VideoINR (Chen et al., 2022b), an important contemporary INR based method in Video Super-Resolution in Appendix A.9.

Table 3: Quantitative comparisons for ×4 Super-Resolution

| Method | PSNR (dB) ↑ | |
|--------|-------|--------|
| | Bunny | Beauty |
| Bicubic | 29.82 | 34.03 |
| ZSSR | 27.53 | 31.96 |
| SIREN | 21.68 | 29.61 |
| **Ours** | **31.74** | **36.48** |

Figure 6: Qualitative Results for ×4 Super-Resolution. PNeRV's superior performance can be observed in the high frequency regions.

Table 4: PSNR (dB) metrics for Video Frame Interpolation

| | Seen Frames | | Unseen Frames | |
|--------|-------|--------|-------|--------|
| Method | Bunny | Beauty | Bunny | Beauty |
| NeRV-L | 39.3 | 36.16 | 28.58 | 23.98 |
| E-NeRV | 42.52 | 36.96 | 33.77 | 26.41 |
| **Ours** | **43.10** | **38.66** | **33.91** | **28.63** |

Table 5: PSNR (dB) metrics for Video Denoising

| | Type of Noise | |
|--------|-------|---------------|
| Method | white | salt & pepper |
| NeRV-L | 38.41 | 39.83 |
| E-NeRV | 37.73 | 38.98 |
| **Ours** | **39.62** | **41.89** |

### 4.3 Video Frame Interpolation

The temporally continuous nature of PNeRV, allows us to perform the task of Video Frame Interpolation. We train and evaluate PNeRV on the "Bunny" and "Beauty" videos for this task. We report the quantitative and qualitative comparisons for the task in Table 4 and Fig. 10 (Appendix A.6) (right column), respectively. We observe that our method achieves better metric performance than prior art, and excellent perceptual quality of the predicted "unseen" (interpolated) frames. Hence, we infer that PNeRV better captures spatiotemporal correlations in videos with respect to prior art. We present additional results for the task in Appendix A.11.

#### 4.3.1 Video Denoising

INRs have been shown to be better attuned to filtering out inconsistent pixel intensities i.e. noise and perturbations. Hence, making it suitable for denoising videos without being explicitly trained for the task. To test the performance of our representation on noisy videos, we applied white noise and salt and pepper noise separately to the original videos. PNeRV was then trained on these perturbed videos for reconstruction. Comparisons between the reconstructed videos and the original videos reveal that the representation learned by PNeRV is robust to noises. It implicitly learns a regularization objective to filter out noise better than existing methods. Quantitative comparisons with prior art (reported in Table 5) assert the superiority of our method. We also provide qualitative results and a detailed analysis of the same in Appendix A.10.

### 4.4 Ablation Studies

#### 4.4.1 Varying the polynomial attributes of the INR Decoder

We study the impact of varying the *rank* and *order* of the polynomial formed by the PNN-based INR Decoder architecture.

**Rank of the Polynomial:** In NCP-Polynomial formulation, the rank of the polynomial can be varied by modifying the number of channels of the $F_{rm}$ module in each ProdPoly block. In general, it is expected that a polynomial with a higher-ranked decomposition (i.e. more channels) would perform better due to the increased expressivity of the representation learned by the model. To understand the effect of this, we modify the rank of the first ProdPoly block in the INR-Decoder while keeping the ranks of the second and third ProdPoly block fixed. These results are reported in Tab. 6. It can be seen that the rank of the polynomial is positively correlated to the quality of the reconstructed video.

**Order of the Polynomial:** Each ProdPoly block in the proposed architecture has an order of 2. Thus, the effective order of INR-Decoder is $2^R$ where $R$ is the total number of ProdPoly blocks in the decoder.

Table 6: **Ablation:** Effect of variation of the rank (controlled by the number of channels) of individual ProdPoly decompositions in terms of PSNR with respect to reconstruction on "bunny" video.

| Rank of the Polynomial Component | | | PSNR (dB) |
|---|---|---|---|
| **ProdPoly: 1** | **ProdPoly: 2** | **ProdPoly: 3** | |
| 324 | 96 | 96 | 44.9 |
| 212 | 96 | 96 | 42.05 |
| 112 | 96 | 96 | 39.23 |

Table 7: **Ablation:** Effect of variation of the order (controlled by the number of ProdPoly blocks) in terms of PSNR for reconstruction on "bunny" video.

| # ProdPoly Blocks | # Params | PSNR |
|---|---|---|
| 2 | 11.50 M | 43.78 |
| 3 | **11.89 M** | **44.90** |
| 4 | 12.29 M | 44.65 |

Table 8: **Ablation:** Effect of the individual PE components formulation on PSNR (dB) for reconstruction.

| Setup | $\Gamma_{PPE}(.)$ | $\Gamma_{HMF}(.)$ | $\Gamma_{TSE}(.)$ | Bunny | Beauty |
|---|---|---|---|---|---|
| Baseline | - | - | - | 41.85 | 35.34 |
| $\lambda_{ij} = Centroid(P_{ij})$ | - | - | - | 42.09 | 39.06 |
| Parametric PE only | ✓ | ✓ | ✗ | 43.83 | 39.70 |
| **Ours** | ✓ | ✓ | ✓ | 44.9 | 39.8 |

Table 9: **Ablation:** Characterizing the effect of varying patch sizes in terms of #Parameters and PSNR (dB) for reconstruction.

| Patch-size | # Parameters | Bunny | Beauty |
|---|---|---|---|
| **H**/8, **D**/8 | 12.37 M | 42.02 | 37.21 |
| **H**/4, **D**/4 | **11.89 M** | **44.90** | **39.80** |
| **H**/2, **D**/2 | 12.56 M | 44.27 | 33.82 |
| **H**, **D** | 12.94 M | 42.65 | 32.46 |

Hence, we vary the number of ProdPoly blocks to change the order of INR-Decoder polynomial and report our findings in Table 7. It can be seen that the performance drops when the order is reduced. Interestingly, the PSNR value decreases when the order is increased beyond a certain range. We also present an analysis of PNeRV's independence to the choice of non-linear activations in Appendix A.12, a property it inherits from the PNN paradigm.

### 4.4.2 Efficacy of Positional Embeddings

We demonstrate the contribution of each Positional Embedding (PE) with respect to its individual contribution toward the reconstruction quality achieved. To this end, we first propose two simple baselines as shown in Table 8 wherein each patch is assigned a coordinate from 0 to $M \times N - 1$ in a row-wise fashion (row 1) or each patch is assigned its centroid value (row 2). Then $\Gamma_{FPE}$ is used to compute the patch embeddings. It is evident that the performance drops considerably in both these settings. Hence, motivating the need of carefully designed positional embeddings. Next, we add the parametric PE ($\Gamma_{PPE}$) (row 3) followed by addition of functional PE ($\Gamma_{TSE}$). The results show that both $\Gamma_{PPE}$ and $\Gamma_{TSE}$ contribute to the overall network performance. For this ablation study, $t$ is encoded using $\Gamma_{FPE}$ and fused with the spatial embedding using the HMF block in all the experiments. Our well-designed PE scheme greatly enhances our model's performance by leveraging the high-frequency information preferred by the PNN paradigm.

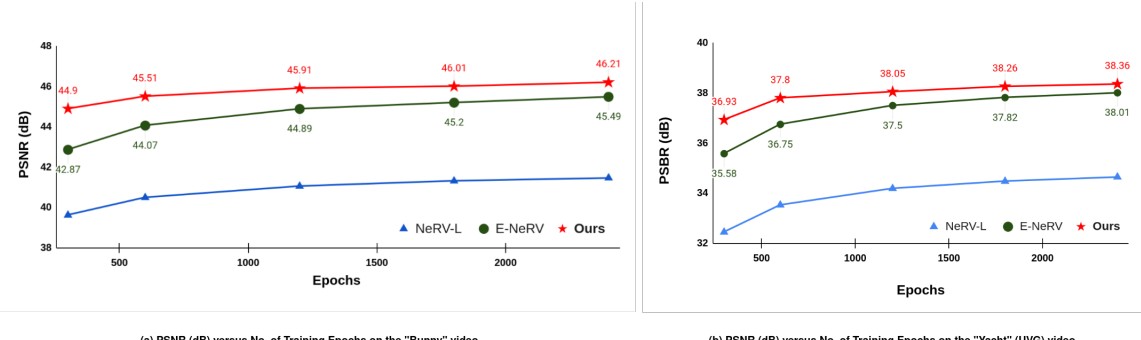

(a) PSNR (dB) versus No. of Training Epochs on the "Bunny" video.    (b) PSNR (dB) versus No. of Training Epochs on the "Yacht" (UVG) video.

Figure 7: Rate of Convergence (PSNR (dB) for reconstruction versus #training epochs) compared to state-of-the-art.

### 4.4.3 Varying the Input Patch-Size

The patch-wise formulation is the key idea that enables us to model spatial continuity. Thus, we delve into PNeRV's performance obtained for different patchs sizes in Table 9. We found that a patch size of $(\frac{\mathbf{H}}{4}, \frac{\mathbf{D}}{4})$ performs the best. This suggests that neither a pixel-wise (dense spatial) nor frame-wise representation (temporal-only) is optimal. We hypothesize that the surge in parameters (over-parameterization) in the pixel-wise approach might be the limiting factor that inhibits learning in such cases. We find this result particularly insightful since we found a sweet-spot between the two parameterization methodologies. Another interesting trend to observe from Table 9 is that of the relationship between patch size and the number of parameters. We discuss the reasoning behind this in detail in appendix A.13.

### 4.4.4 HMF versus other fusion strategies

We compare the proposed PNN-backed Higher-order Multiplicative Fusion (HMF) of space and time embeddings with other fusion mechanisms as given in Table 10. As expected, conventional concatenation, addition, or multiplication operations on features fail to capture the auto and cross-correlations of the inputs. Hence, causing a drop in performance. We observe that the dip in PSNR is more pronounced for the "bunny" video than the "beauty" video. We attribute this observation to the "bunny" video having more temporal variations. The results of this study indicate that the proposed HMF scheme models both the structural and the perceptual video attributes better than the prior art.

Table 10: **Ablation:** Assessing the efficacy of our HMF versus other parametric PE fusion strategies in terms of PSNR (dB) for reconstruction.

| PE Fusion Strategy | Bunny | Beauty |
|---|---|---|
| Concat + Linear | 43.76 | 39.39 |
| Linear + Elementwise Addition | 43.28 | 39.39 |
| Linear + Hadamard Product | 43.06 | 38.92 |
| **Ours** | **44.9** | **39.80** |

### 4.5 On PNeRV's rate of convergence

Following E-NeRV's setup, we perform reconstruction experiments with PNeRV models trained for different number of training epochs on the "Bunny" and "Yacht" (UVG dataset) videos and report our findings in Fig. 7. It can be seen that training for more number of epochs boosts the performance with upto $4\times$ faster convergence than baselines. PNeRV's performance surpasses that of the baselines at 600 epochs on the "Bunny" and 1200 epochs on the "Yacht". We also provide comparisons with state-of-the-art with respect to inference time in Appendix A.8.

## 5 Conclusion

In this work, we propose and validate the efficacy of PNeRV, a light-weight, spatiotemporally continuous, fast, and generic neural representation for videos with a versatile set of practical downstream applications. We do so by building on two principal insights. First, a well-designed patch-wise spatial sampling scheme can perform just as as good as a pixel-wise sampling. Second, replacing popular function approximators by the more efficient PNNs and designing other model components to aid its learning can lead to superior performance. We provide conclusive results to support our claims with analysis on several downstream tasks and consistent ablation studies. We believe our work shall serve as a primer toward building spatiotemporally continuous light-weight INRs for videos. As a future work, it would be interesting to examine PNN based PEs to further improve INR for videos. Please find our broader impact statement in the following subsection.

### 5.1 Broader Impact Statement

As one of the most widely consumed modality of data, videos are central to several important tasks in the modern socio-technical context. In such a scenario, PNeRV brings in a fresh approach to tackle the ever growing costs involved in handling such massive data by providing a method restore and compress videos efficiently. In effect, PNeRV can potentially have a lasting positive impact on several video streaming, communication, and storage services. As with any nascent technology, the largely positive impact areas are accompanied by a few unforeseeable ones which are beyond the scope of this work.

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

# A Appendix

## A.1 Abbreviations

| | |
|---|---|
| **INR** | Implicit Neural Representation |
| **PNN** | Polynomial Neural Network |
| **HMF** | Higher-order Multiplicative Fusion |
| **PE** | Positional Embedding |
| **FPE** | Positional Encoding of Frame Index |
| **PPE** | Parametric Embedding of Fine Coordinates |
| **TSE** | Time Aware Spatial Embedding |

## A.2 The mode-n product

The mode-$n$ (matrix) product of a tensor $\mathcal{X} \in \mathbb{R}^{I_1 \times I_2 \times \dots \times I_N}$ with a matrix $\mathbf{U} \in \mathbb{R}^{J \times I_n}$ is denoted by $\mathcal{X} \times_n \mathbf{U}$ and is of size $I_1 \times \dots I_{n-1} \times J \times I_{n+1} \times \dots \times I_n$. Elementwise, we have

$$(\mathcal{X} \times_n \mathbf{U})_{i_1 \dots i_{n-1}} j i_{n+1} \dots i_N = \sum_{i_n=1}^{I_n} x_{i_1 i_2 \dots i_N} u_{j i_n}.$$

Each mode-$n$ fiber [of $\mathcal{X}$] is multiplied by the matrix $\mathbf{U}$.

## A.3 The Hierarchical Patch-wise Spatial Sampling Algorithm

---
**Algorithm 1** Hierarchical Patch-wise Spatial Sampling

---
**Input:** $\mathcal{C}$, $H$, $D$, $K$, $L$, $M$, $N$, $\boldsymbol{P_{ij}}$, $\tilde{\boldsymbol{P}}_{\boldsymbol{kl}}$
**Output:** $\lambda_{ij}, \boldsymbol{\Lambda_{ij}}$

1: **function** HPSS($\mathcal{C}, \boldsymbol{P_{ij}}, \tilde{\boldsymbol{P}}_{\boldsymbol{kl}}, H, D, K, L, M, N$)
2:      $\lambda_{ij} \leftarrow (\lfloor \frac{top(\boldsymbol{P_{ij}})+bottom(\boldsymbol{P_{ij}})}{2} \rfloor, \lfloor \frac{left(\boldsymbol{P_{ij}})+right(\boldsymbol{P_{ij}})}{2} \rfloor)$
3:      $\boldsymbol{\Lambda_{ij}} \leftarrow$ A matrix of dimensions $K \times L$ with $k$ and $l$ being the row and column index, respectively.
4:      $x = top(\tilde{\boldsymbol{P}}_{\boldsymbol{kl}})$, $y = left(\tilde{\boldsymbol{P}}_{\boldsymbol{kl}})$, $h = \frac{H}{MK}$, $d = \frac{D}{NL}$
5:      **for** $k \leftarrow 0$ to $K-1$ **do**
6:          **for** $l \leftarrow 0$ to $L-1$ **do**
7:              $\boldsymbol{\Lambda_{ij}}[k][l] \leftarrow (\lfloor \frac{2x+h}{2} \rfloor, \lfloor \frac{2y+h}{2} \rfloor)$
8:              y = y+h
9:          **end for**
10:        $y = left$
11:        $x = x + d$
12:      **end for**
13:      **return** $\lambda_{ij}, \boldsymbol{\Lambda_{ij}}$
14: **end function**

---

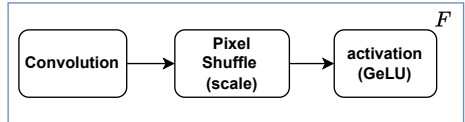

Figure 8: Detailed architecture of the $F$ blocks.

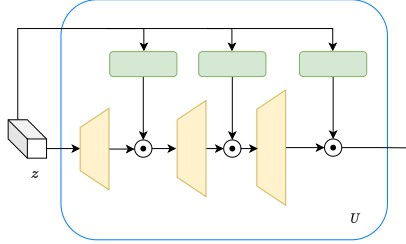

Figure 9: Detailed diagram for the $U$ block in the INR Decoder. Yellow blocks represent the transpose convolutional layers, whereas the green rectangles are fully connected layers.

| Layer | Modules | Upscale Factor | Output Size $C \times H \times W$ |
|---|---|---|---|
| U | MLP & TransposeConv2D & Reshape | - | $324 \times 16 \times 9$ |
| $\Phi_{11}$ | MLP | - | $160 \times 324$ |
| $F_{11}$ | F block | 5 | $324 \times 80 \times 45$ |
| $\Phi_{12}$ | MLP | - | $160 \times 162$ |
| $F_{12}$ | F block | 2 | $162 \times 160 \times 90$ |
| $\Phi_{21}$ | TransposeConv2D | 2 | $384 \times 320 \times 180$ |
| $F_{21}$ | F block | 2 | $384 \times 320 \times 180$ |
| $\Phi_{22}$ | TransposeConv2D | - | $96 \times 320 \times 180$ |
| $F_{22}$ | F block | - | $96 \times 320 \times 180$ |
| $\Phi_{31}$ | TransposeConv2D | - | $96 \times 320 \times 180$ |
| $F_{31}$ | F block | - | $96 \times 320 \times 180$ |
| $\Phi_{32}$ | TransposeConv2D | - | $96 \times 320 \times 180$ |
| $F_{32}$ | F block | - | $96 \times 320 \times 180$ |
| ToRGB Layer | Convolution | - | $3 \times 320 \times 180$ |

Table 11: INR-Decoder Architecture.

## A.4 The INR Decoder architecture in detail

In this section, we provide the finer details of the PNeRV architecture. We then provide more details about the implementation and training of the proposed method. PNeRV consists of three components: the Positional Embedding Module (PE), the Embedding Fusion Block, and the INR-Decoder. Given the coarse patch coordinate $\lambda_{\mathbf{ij}}$, fine patch coordinate $\mathbf{\Lambda}_{ij}$ and the time index, we first compute the positional embeddings $\mathbf{\Gamma}_{TSE}(\lambda_{\mathbf{ij}}, t)$, $\mathbf{\Gamma}_{PPE}(\mathbf{\Lambda}_{ij})$ and $\mathbf{\Gamma}_{FPE}(t)$. The embeddings $\mathbf{\Gamma}_{PPE}(\mathbf{\Lambda}_{ij})$ and $\mathbf{\Gamma}_{FPE}(t)$ are fused using a Polynomial Neural Networks (PNN) based fusion module HMF. HMF consists of a series of linear transformations followed by Hadamard product and addition, as shown in Fig 4 of the paper. Each linear layer, namely, $A_{[1,t]}, A_{[2,t]}, A_{[3,t]}, A_{[1,\mathbf{\Lambda}_{ij}]}, A_{[2,\mathbf{\Lambda}_{ij}]}, A_{[3,\mathbf{\Lambda}_{ij}]}$ is of dimension $80 \times 160$ . The resulting embedding is added elementwise to $\mathbf{\Gamma}_{TSE}(\lambda_{\mathbf{ij}}, t)$ to obtain the fused embedding $z$ which is given as input to the INR Decoder. $z$ is a vector of dimension 160.

The INR-decoder consists of a stack of 3 prodpoly blocks. Each prodpoly block in turn is a 2$^{\text{nd}}$ order NCP-Polynomial implemented using convolutional blocks $F_{rm}$, where $r$ is the index of the prodpoly block and $m$ is the index corresponding to the F-block. The structure of $F$ is illustrated in Fig. 8. To limit the increase in the number of parameters of the model, following (Li et al., 2022b), we employ the following design for $F_{11}$ block: $\text{Conv}(C_1, C_0 \times s \times s) \to \text{pixel-shuffle}(s) \to \text{Conv}(C_0, C_2)$. Where, $C_1 = 324, C_0 = 81, C_2 = 324$ and $s = 5$. The input vector $\boldsymbol{z}$ is mapped to a feature map using a 3rd-order polynomial implemented using transpose convolutional layers as depicted in Fig. 9. This is referred to as $U$ in Fig. 2. Table. 11 provides the complete architecture details for INR-decoder.

### A.5    Qualitative results for Video Reconstruction

We provide additional comparisons with state-of-the-art in Fig. 16 and additional qualitative results for our method illustrated in Fig. 14 and Fig. 15. Owing to the ensemble of design elements, PNeRV outperforms state-of-the-art convincingly on this task.

### A.6    Qualitative Comparisons

Figure 10 (Appendix A.6) presents qualitative comparisons with prior art on Reconstruction, Compression, and Interpolation.

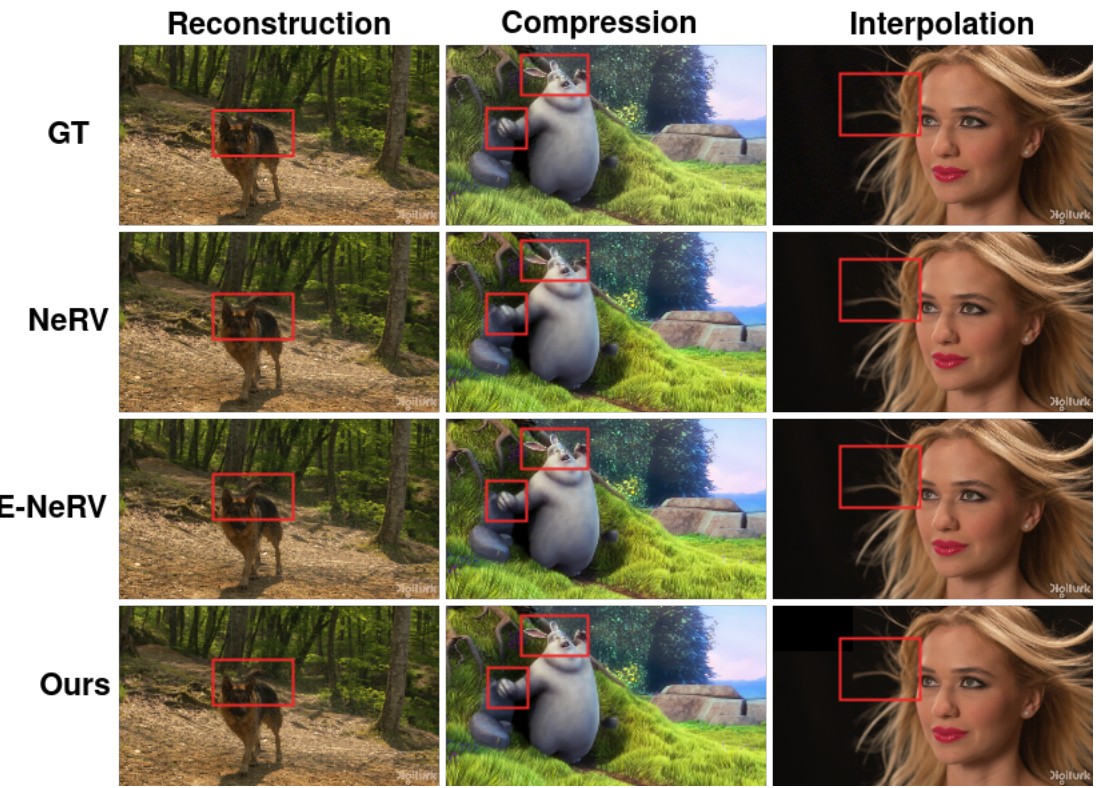

Figure 10: Qualitative comparisons with prior art on Reconstruction, Compression, and Interpolation. The specific regions where our method predicts significantly better outputs are highlighted in red boxes.

### A.7    Additional Results for Video Compression

Table 12 (a) provides a quantitative comparison with state-of-the-art on video compression in different sparsity (denoted by $\rho$) settings. Our model outperforms prior art convincingly. Fig. 17 wherein we present qualitative comparisons with state-of-the-art on the task with sparsity $\rho = 0.2$, further underscores PNeRV's superior performance.

Table 12: (a) **Averaged Comparison -** Model Compression (b) **Quantitative Comparison -** Inference Time.

| Method | (a) Compression PSNR (dB) ↑ | | (b) Inference Speed |
|---|---|---|---|
| | $\rho = 0.2$ | $\rho = 0.4$ | Inference Time (ms) ↓ |
| NeRV | 32.30 | 31.20 | 153.81 |
| E-NeRV | 32.54 | 32.28 | 34.11 |
| **Ours** | **33.86** | **33.60** | **28.32** |

### A.8    Quantitative Comparison: Inference time per forward pass

Table 12 (b) provides a quantitative comparison with state-of-the-art in terms of time taken (ms) to perform one forward pass of the model on NVIDIA GeFORCE RTX 3090 GPU. Results elucidate that our light-weight model is faster then prior art.

### A.9    Super-Resolution using PNeRV

**On the comparison with VideoINR:**   VideoINR (Chen et al., 2022b) has two core differences from our work. Firstly, VideoINR uses ground truth High-Resolution (HR) video frames for training, while ours is a fully unsupervised approach utilizing only the low-resolution video for training. Secondly, our method is a multifunctional INR. In that, it learns to represent a signal (video) as model weights. In contrast, VideoINR is an autoencoder trained specifically for Super-Resolution. Wherein, the claimed INR components function as non-linear transformations in the intermediate feature space. Therefore, we do not compare with VideoINR. Instead, we show qualitative results for Super-Resolution (Fig. 6) and quantitative comparison with bicubic interpolation, ZSSR, and SIREN (Table 4) which are unsupervised models.

### A.10    Video Denoising: Qualitative Results

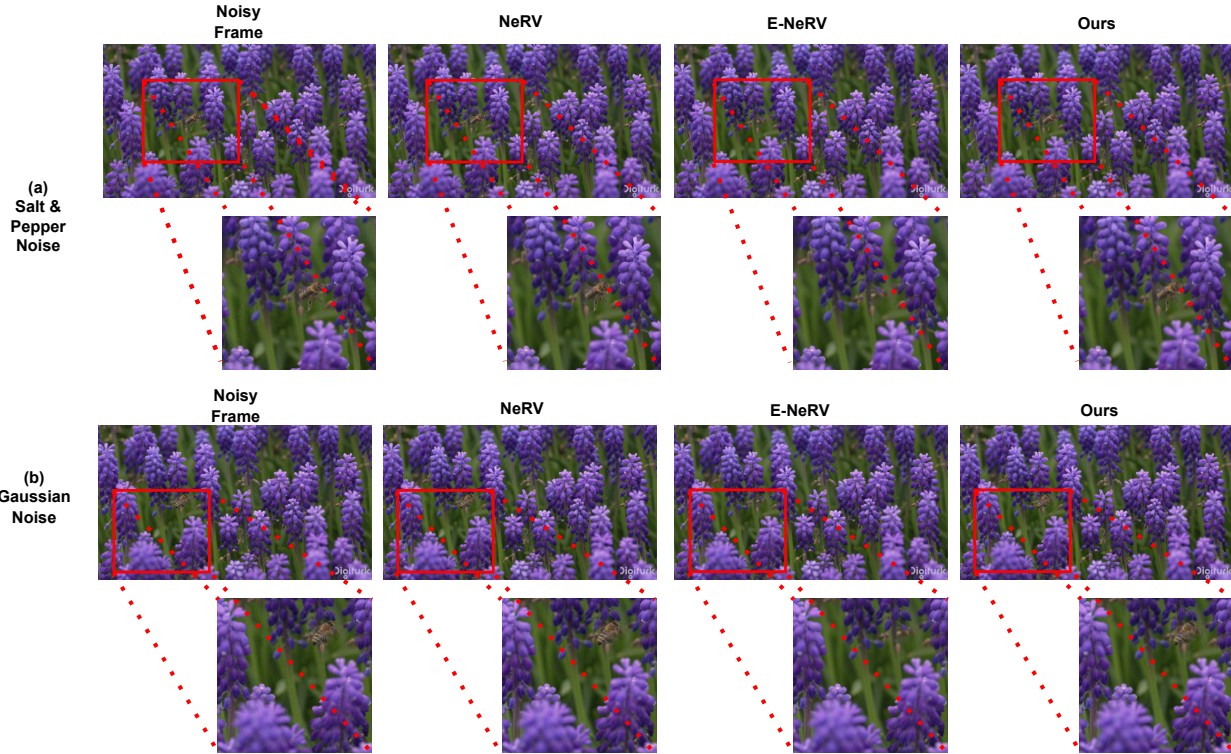

Figure 11: Qualitative Comparison of denoising results obtained on "honeybee" video. (a) Salt and Pepper Noise, (b) Gaussian Noise.

Figure 11 shows the qualitative comparison of the output of our method with the two INR baselines NeRV (Chen et al., 2021) and E-NeRV (Li et al., 2022b). Notice that E-NeRV fails to reconstruct the honeybee, thus regularizing the video such that the original content is lost. NeRV can generate the honeybee but it lacks clarity. PNeRV preserves all the content of the frames including honeybee and generates superior-quality video. These results confirm that PNeRV learns more robust video representation.

Table 13: Metrics to analyze the effect of absence of non-linear activations in terms of PSNR (dB) for reconstruction.

| Method | Bunny | Beauty |
|--------|-------|--------|
| NeRV-L | 31.71 | 27.53 |
| E-NeRV | 27.57 | 27.49 |
| **Ours** | **39.84** | **38.50** |

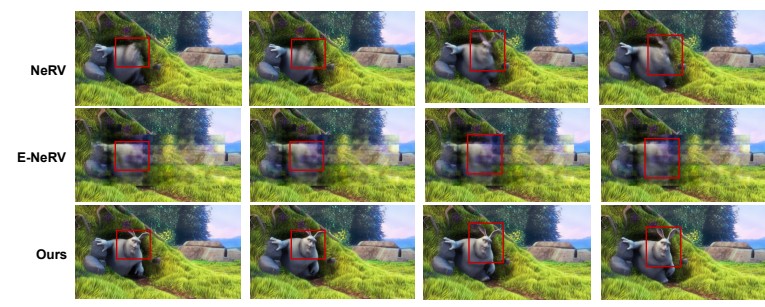

Figure 12: Visualization of frames reconstructed by models trained without non-linear activation functions. The highlighted regions illustrate our method's robustness to the choice activation employed.

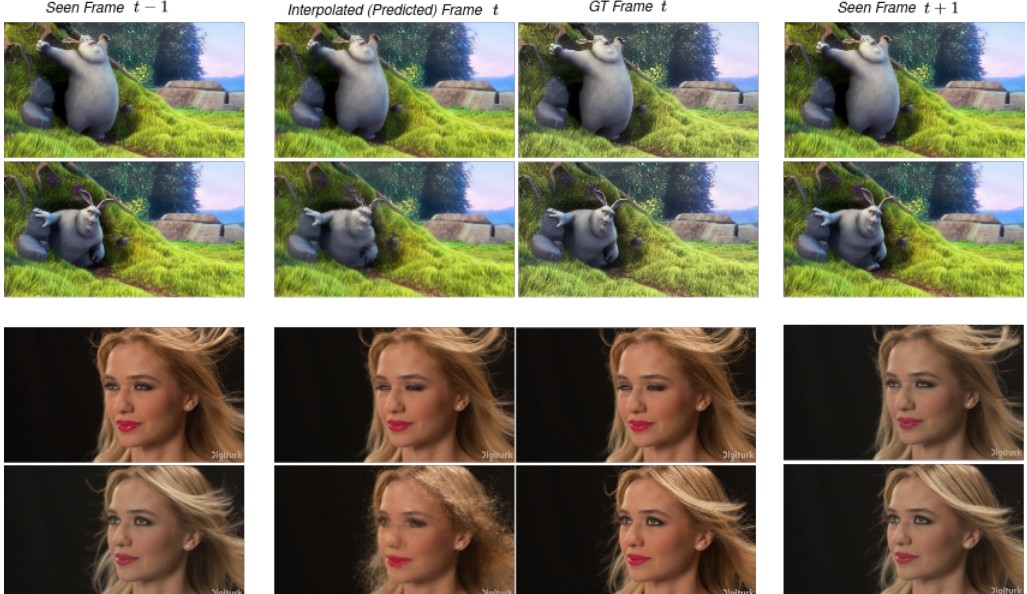

Figure 13: Qualitative results for Video Frame Interpolation on the "Bunny" (rows 1 and 2) and "Beauty" (rows 3 and 4) videos. Columns 1 (seen, previous) and 4 (seen, next) show the seen frames used to interpolate (predict) the unseen frame illustrated in column 2. The closeness of predicted frames (column 2) to the ground truth frames (column 3) underscores the faithfulness of our interpolation.

## A.11 Video Frame Interpolation

Following E-NeRV's setup, we divide the training sequence in a 3:1 ("seen:unseen") ratio such that for every four consecutive frames, the fourth frame is not used training. This "unseen" frame is interpolated during inference to quantitatively evaluate the model's performance.

Table 14: MS-SSIM metrics for Video Frame Interpolation

| Method | Beauty |
|--------|--------|
| NeRV-L | 0.8161 |
| E-NeRV | 0.9745 |
| **Ours** | **0.9775** |

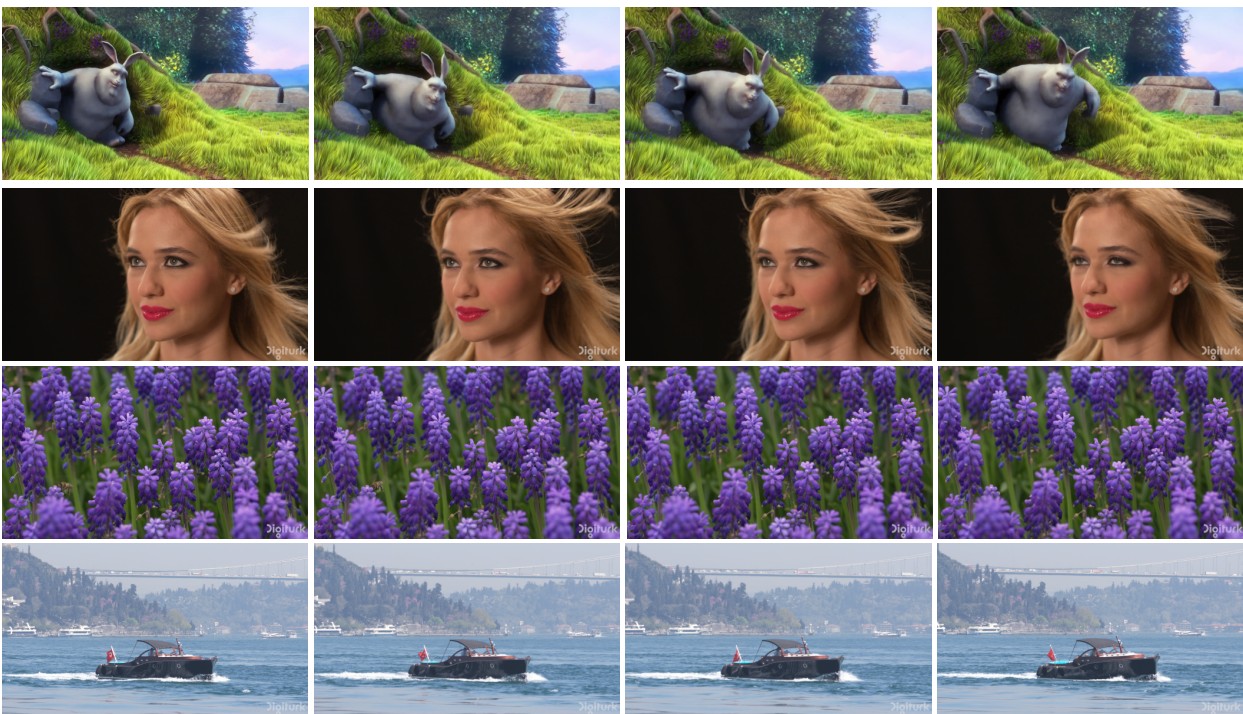

Figure 14: Visualization of few frames of the reconstructed videos on "bunny" (first row), "beauty" (second row), "honeybee" (third row) and "bosphorus"(fourth row) videos of UVG dataset (Mercat et al., 2020).

Fig. 13 provides the qualitative results for Video Frame Interpolation task on "bunny" and "beauty" videos. It can be observed that the perceptual quality of the interpolated frame is similar to that of the ground truth for the bunny video. We also assess the quality of the interpolated video using multi-scale structural similarity (MS-SSIM) metric. MS-SSIM accounts for luminance, contrast and structure of each frame and hence better correlates with human perception. Table 14 reports the MS-SSIM numbers on "beauty" video. PNeRV outperforms SOTA methods on this metric as well suggesting the superior quality of the interpolated video.

### A.12 Robustness to the choice of activation function

Since PNNs (Chrysos et al., 2021b) have built-in non-linearities, they do not rely on the usage of popular hand-crafted non-linear activation functions to yield best performance. To highlight this aspect of our method, we test the effect of training our network and the baselines without any activation functions on "bunny" dataset by removing activation functions from all the network layers except for the output layer. The quantitative and qualitative results for the same are reported in Table 14 and Fig. 12. It can be observed that performance of the baselines NeRV (first row) and E-NeRV (second row), dropped significantly. In contrast, the performance of our model remains comparable. It is notable that NeRV fails to learn high-frequency information such as that in the face of the bunny (highlighted in red boxes), resulting in a worse qualitative performance.

### A.13 Relationship between patch-size and number of learnable parameters

Intuitively, one would expect that number of learnable parameters would increase with an increase in patch size. However, as observed in Table 9, the number of parameters decreases when transitioning from a patch size of $(\frac{H}{8}, \frac{W}{8})$ to $(\frac{H}{4}, \frac{W}{4})$ because of a lesser number of channels in convolutional layers of the prodpoly blocks for the latter.

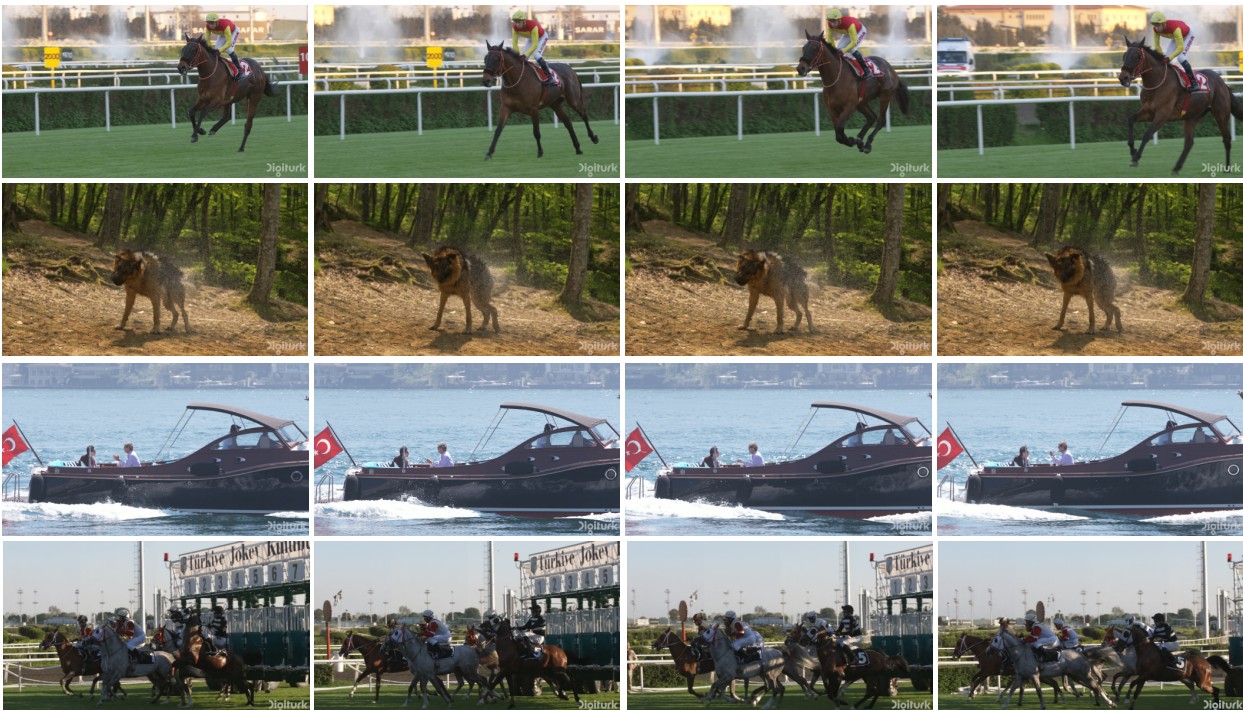

Figure 15: Visualization of few frames of the reconstructed videos on "jockey" (first row), "shakeandry" (second row), "yachtride" (third row) and "readysetgo" (fourth row) videos of UVG dataset (Mercat et al., 2020).

To be specific, for $(\frac{H}{8}, \frac{W}{8})$ configuration, the number of channels in $F_{22}$, $F_{31}$ and $F_{32}$ (see Table 11 in Appendix A.3 and Figure 2) are 384 whereas for the $(\frac{H}{4}, \frac{W}{4})$ scenario, the number of the channels are all set to 96. We arrive at this design choice empirically. The F blocks comprise pixel shuffle layers which are responsible to upsample the size of the generated output. In our design, the number of channels are chosen to be as 96, or channels in the previous layer divided by the upsampling factor square, whichever is minimum. For patch sizes greater than or equal to $(\frac{H}{8}, \frac{W}{8})$, the channels in the $F_{22}$, $F_{31}$ and $F_{32}$ layers become 96. The increase in parameters after these layers is due to the $\Psi_{rm}$ layers that upsample the input feature map to appropriate dimensions. This contributes to the extra parameters.

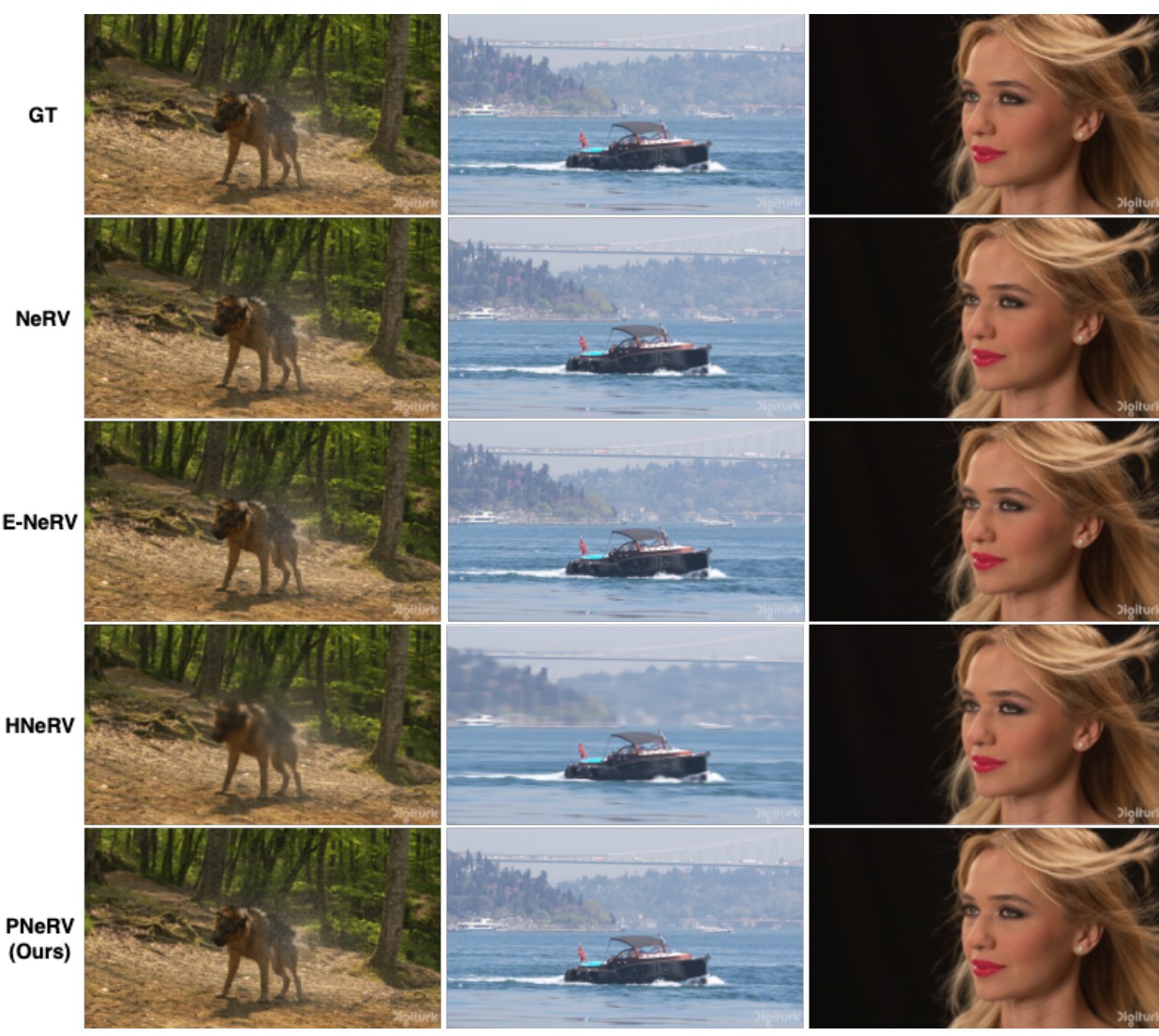

Figure 16: Qualitative comparisons with state-of-the-art with respect to video reconstructing on the "shake" (column 1), "bosphorus" (column 2), and "beauty" (column 3) videos of the UVG dataset (Mercat et al., 2020). "GT" denotes the ground truth frames. As is evident, PNeRV outperforms state-of-the-art, particularly in regions with high frequency information content.

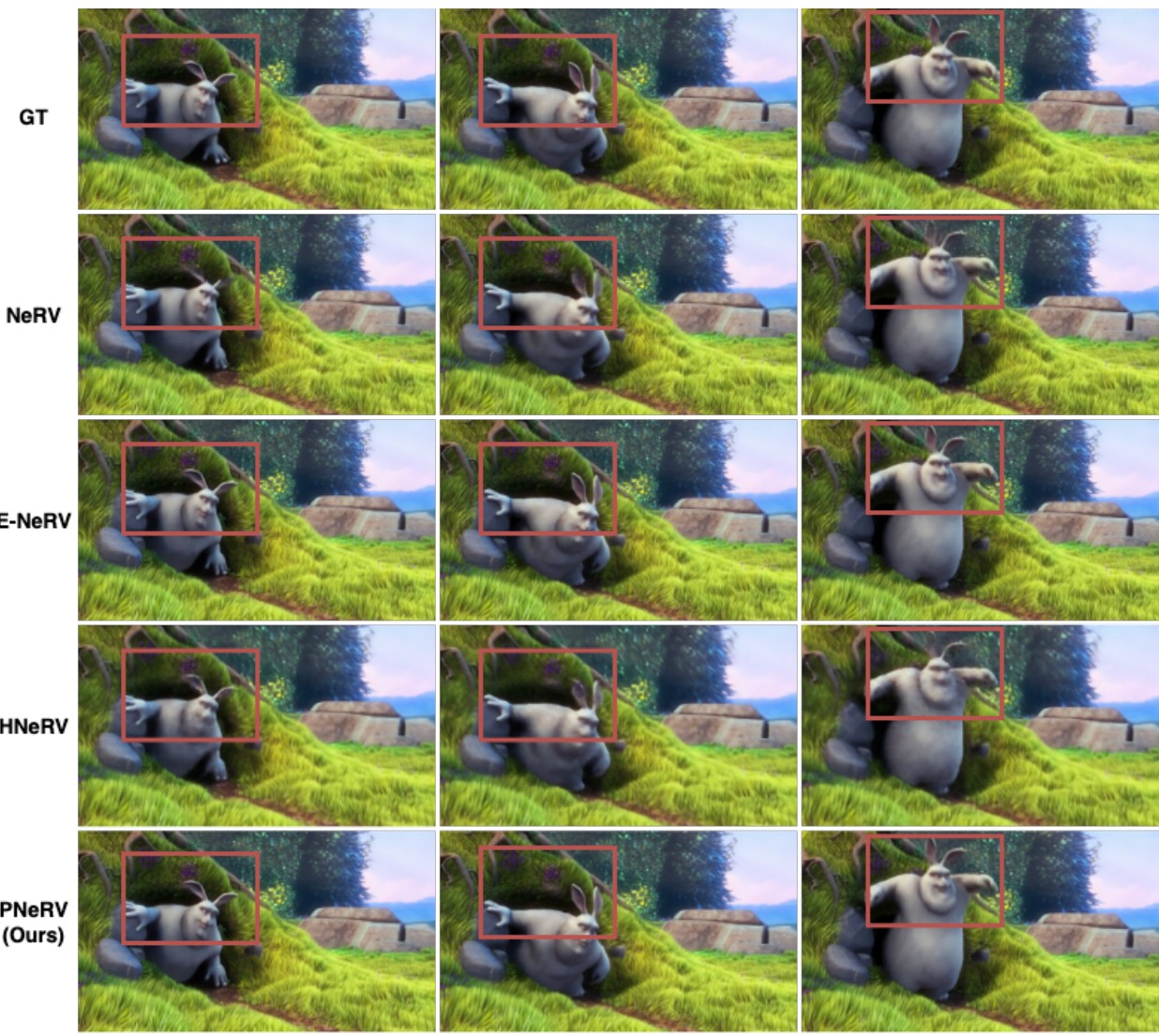

Figure 17: Qualitative comparison with state-of-the-art with respect to video compression ($\rho = 0.2$) on the "Bunny" video. "GT" denotes ground truth frames. The highlighted regions depict regions where PNeRV outperforms state-of-the-art evidently.

