# OpenReview forum: "PNeRV: A Polynomial Neural Representation for Videos"
_TMLR — Accepted by TMLR_

### Review · Reviewer_65SM · 2023-10-19

**Summary Of Contributions:**

The paper proposes a novel neural representation (NR) for videos based on two principals: patch-wise spatial sampling that enables modeling spatial continuity while ensuring fast inference and leveraging Polynomial Neural Network (PNN)s for modeling high resolution videos with fewer parameters. The proposed NR, Polynomial Neural Representation for Videos (PNeRV), is equipped with the new positional embeddings (PEs) that utilizes both parametric and functional PEs, which enhances the model's ability to capture the high frequency information and both the auto and cross-correlations of the inputs. The experimental results on video reconstruction and other downstream tasks (video compression, video super-resolution and video denoisig) showcase the superiority of the proposed model compared to previous NR approaches.

**Audience:**

Yes

**Broader Impact Concerns:**

The authors adequately addressed the broader impact of the work in the paper.

**Claims And Evidence:**

Yes

**Requested Changes:**

Please fix or explain the weaknesses I mentioned above.

**Strengths And Weaknesses:**

Strengths
+ The overall model architecture is explained in detail
+ The design choices that the authors made, including how to fuse positional embeddings and the architecture of INR decoder, improved the model's performance over the baselines, which was proved by the extensive experimental results and ablation studies.

Weakness
Overall, the writing and presentation should be improved.
- There are many typos in equations and tables. For example, where does H in the first paragraph on page 7 come from? The HMF contains addition between x_i and x_{i+1} according to Figure 4 while Eq. 8-9 do not. The first paragraph on page 11 says that the higher-ranked decomposition (more channels) perform better while Table 6 contains the opposite.
- The paper lacks some important details for understanding the content. For example, why does the number of parameters decrease and then increase again when the patch size decreases? (Table 9)

---

### Review · Reviewer_LSEQ · 2023-11-16

**Summary Of Contributions:**

- The authors propose a novel network architecture for neural representation for videos. The architecture consists of the following components:
	- 1. A positional embedding module, the authors combine Fourier features, angle modulation and fusion.
	- 2. A polynomial neural network as the backbone, instead of a "conventional" neural implicit network.
	- 3. A Hierarchical sampling scheme, consisting of coordinates for coarse and finegrained samples.

**Audience:**

Yes

**Claims And Evidence:**

Yes

**Requested Changes:**

- Writing
	- The paper contains many unusual formulations, please carefully proofread the paper to find and fix them. Examples:
		- "such as **non-reliance** on the employment"
		- "We further propose a PNN based Higher order Multiplicative Fusion (HMF) module that is responsible for learning parametric embedding."
		- "In contrast, PNeRV **gravitates** back to a spatiotemporal parameterization"
		- "The HPSS methodology is **encapsulated** by Algorithm 1 in Appendix A.2."
	- Many different abbreviations are introduced (HMF, PPN, PNN, INF, PE, TSE, PPE, EFB) and used instead of the original nouns in the text. This makes many sentences very hard to read. Please use these abbreviations sparingly.
- Formatting
	- Text in tables is way too small
	- Weird parts of the text are in red. (around eq. 8, Table 6. and Tabl 9.)
	- PSNR is crossed in Table 9. What is this supposed to indicate?
	- Please use larger images in Figure 5. These images are not suitable for qualitative inspection at all. If in need for space, only show specific crops of the image.
- Questions about validity of the Hierarchical Sampling scheme.
	- As can be seen in equation 3, the proposed model outputs a patch $P_{ij}$ based on coarse $\Lambda$ and fine $\lambda$ input coordinates. It is not clear to me what the hierarchical component of this scheme is. Given that both $\Lambda$ and $\lambda$ represent different frequencies of the single patch centroid, any conventional neural implicit representation should include $\Lambda$ already as a low-frequency embedding of $\lambda$ .
	- The authors should make more clear in the text, or even with experiments what is the added value of their "hierarchical" sampling scheme because it is not clear to me.
	- In paragraph 3.1 the authors contrast this with a naïve path-index. I think this is not a realistic baseline since neural implicit representations always map embedded 2d coordinates to the desired output.
- Questions about the complexity of the embedding scheme.
	- The authors propose a very complicated coordinate embedding scheme. They are somewhat motivated by the ablation results in Table 8. But I wonder if there are not simpler methods with the same performance.
- Ablation on the key contributions.
	- The paper seems to consist of three main components:
		- 1. A positional embedding module, the authors combine Fourier features, angle modulation (similar to Mai & Liu 2022)
		- 2. A polynomial neural network as the backbone, instead of a "conventional" neural implicit network.
		- 3. A Hierarchical sampling scheme, consisting of coordinates for coarse and finegrained samples.
	- While the authors do provide a multitude of ablations, it would be nice if they could include a single simple ablation that shows the performance of each of these three contributions individually. This would allow me to better judge the contribution to the (good) final performance.
	- As a reference, the E-NeRV paper itself has a very nice example of such an ablation in Figure 3.

**Strengths And Weaknesses:**

- ### Strenghts
	- Novelty of adapting polynomial neural networks for implicit neural representations. Using the hierarchical sampling
	- Results for compression and frame interpolation seem to be good, outperforming previous methods.
	- The method is evaluated on multiple tasks and on the standard benchmarking dataset UVG and on the Big buck bunny Video.
	- Ablation studies
- Weaknesses

	- Writing quality and presentation of the work can use a lot of improvement. (See requested changes below)
	- The methodology seems very complicated. There are a lot of minor engineering details. It is good that the authors described all of these but it makes the paper hard to follow.
	- The validity of the novel embedding scheme and the hierarchical sampling procedure seem questionable to me. I want to clarify that it could be due to me not understand the concepts clearly. Something that the authors could try to address by improving the writing quality.

---

> ### Author Response · Authors · 2023-12-26
> **Response to Reviewer LSEQ (Part 1)**
>
> We are truly thankful to the reviewer LSEQ for their detailed feedback and their assistance in improving our manuscript. We appreciate the reviewer finding the integration of PNNs in INR for videos, novel. We also acknowledge their appreciation of the performance of our method on several downstream tasks. Below, we address one by one the remarks of the reviewer:
>
> **Q1: The paper contains many unusual formulations, please carefully proofread the paper to find and fix them.**\
> **A1:** Thank you for bringing this to our attention. We have proofread the paper and modified such phrases to enhance the clarity of the paper. For instance:
> *  “such as non-reliance on the employment” has been modified to “robustness to the choice of”
> * "We further propose a PNN based Higher order Multiplicative Fusion (HMF) module that is responsible for learning parametric embedding."  has been modified to “ We build a Higher order Multiplicative Fusion (HMF) module that learns parametric embedding.”
> * “In contrast, PNeRV gravitates back to a spatiotemporal parameterization” has been modified to  “In contrast, PNeRV uses a spatiotemporal parameterization”.
> * "The HPSS methodology is encapsulated by Algorithm 1 in Appendix A.2." has been modified to “Algorithm 1 in Appendix A.2 summarizes hierarchical patch-wise spatial sampling.”
>
> **Q2. Many different abbreviations are introduced (HMF, PPN, PNN, INF, PE, TSE, PPE, EFB) and used instead of the original nouns in the text. This makes many sentences very hard to read. Please use these abbreviations sparingly.**\
> **A2:** We acknowledge that too many abbreviations may hinder the readability of the sentences. To that end, we make an effort to improve the readability of our work. Concretely:
> * Acronyms like HMF, PNN, PE and PPE are required as they have been referenced many times in the text, tables and figures.
> * We believe that the acronyms “PPN'' and “INF” are not used in the manuscript. It would be great if the reviewer could kindly point out specifically if they come across the acronyms “PPN” and “INF” in the manuscript.
> * We have removed the use of the acronyms TSE, NR, HPSS, EFB, PEM, NLB, MLP.
> * To enhance the clarity of the paper content, we have defined the abbreviations at the beginning of the appendix as well. Please find the screenshot of the same for quick reference as an anonymous link [here](https://imgur.com/a/jmZq7yK).
>
> **Q3. Ablation on the key contributions:
> The paper seems to consist of three main components:\
> (1) A positional embedding module, the authors combine Fourier features, angle modulation (similar to Mai & Liu 2022)\
> (2) A polynomial neural network as the backbone, instead of a "conventional" neural implicit network.\
> (3) A Hierarchical sampling scheme, consisting of coordinates for coarse and fine grained samples.\
> While the authors do provide a multitude of ablations, it would be nice if they could include a single simple ablation that shows the performance of each of these three contributions individually. This would allow me to better judge the contribution to the (good) final performance. As a reference, the E-NeRV paper itself has a very nice example of such an ablation in Figure 3.**
>
> **A3:** We are thankful to the reviewer for acknowledging the numerous ablations that we have performed. Considering the interdependencies of the proposed components, specifically (1) and (3) i.e. the sampling scheme and the positional embedding used to embed the sampled coordinates, it would not be feasible to include an ablation similar to ENeRV that studies the effect of each component incrementally. This is because, replacing the sampling scheme, would require replacing the positional embedding component as well. Since the previous INRs have not relied on patchwise formulation, unlike ENeRV, we do not have prior art to ensure the alignment between various components of the architecture. Hence, we have provided the ablation of components (1) and (3) in the paper as described below:
> * Positional Embedding Module: Section 4.3.2, table 8 (row 3 and row 4) studies the efficacy of the proposed positional embedding module.
> * Hierarchical Patchwise sampling scheme: Section 4.3.2, table 8 (row 1 and row 2) compares the proposed Hierarchical Patchwise sampling scheme against two possible baselines.
>
> For component 2, i.e. Polynomial Neural Network Backbone, section 4.3.1 studies the performance with different ranks and order of the polynomial backbone. Based on your suggestion we replaced this PNN backbone with an architecture similar to that of NeRV. This achieves a PSNR of 41.70 dB whereas the PNN achieves a performance of 44.90 dB. This suggests that a polynomial backbone plays a crucial role in modeling INR for complex signals like video effectively.

---

> ### Author Response · Authors · 2023-12-26
> **Response to Reviewer LSEQ (Part 2)**
>
> **Q4. Questions about validity of the Hierarchical Sampling scheme.**\
> **a. As can be seen in eq. 3, the proposed model outputs a patch $\boldsymbol{P_{ij}}$ based on coarse $\boldsymbol{\Lambda}$ and fine $\boldsymbol{\lambda}$ input coordinates. What is the hierarchical component of this scheme ? Given that both $\boldsymbol{\Lambda}$ and $\boldsymbol{\lambda}$ represent different frequencies of the single patch centroid, any conventional INR should include $\boldsymbol{\Lambda}$ already as a low-frequency embedding of  $\boldsymbol{\lambda}$.**\
> **A4 (a)** The manner in which the patch-coordinates are sampled in our scheme is hierarchical in nature. Instead of relying on the simple paradigm of sampling all the coordinates of a given patch, we first sample a coarse patch i.e. the centroid of the patch. To bring back the spatial context, we further divide this patch in smaller sub-patches and compute the centroids for each sub-patch. Please note that the coarse patch coordinate $\boldsymbol{\lambda}$ and the fine patch coordinate $\boldsymbol{\Lambda}$ need not be the same. Therefore, the embeddings $\boldsymbol{\lambda}$ and $\boldsymbol{\Lambda}$ can be disjoint.
>
> **b. What is the added value of their “hierarchical” sampling scheme.**\
> **A4 (b)** To highlight the importance of the efficacy of the proposed Hierarchical Patch-wise Spatial Sampling scheme, we design two baselines as illustrated in Table 8. The baseline in row 1, assigns a patch number to each of the patches in a row-wise fashion. Whereas the second baseline (row 2) assigns a 2D coordinate to each patch which is similar to the coarse patch embedding (section 3.1). Row 3 presents the scenario where a fine patch embedding is also used along with the coarse embeddings. The improvement in the performance signifies that both coarse and fine embeddings are required. The intuition behind this is that the coarse coordinate captures a global context whilst the fine coordinates of a patch capture the local context.
>
> **c. In paragraph 3.1 the authors contrast this with a naïve patch-index. I think this is not a realistic baseline since INRs always map embedded 2d coordinates to the desired output.**\
> **A4 (c)** While generating higher-dimensional output from lower dimensional embeddings is a common practice in INR literature, we would like to draw the reviewer’s attention to the fact that the objective of the baseline in question is to assess the efficacy of our patch coordinate sampling strategy and not the other components of the pipeline. The term "sampling strategy" refers to the indexing scheme for patches employed while generating their embeddings. To that end we use two baselines to contrast with our Hierarchical Patch-wise Spatial Sampling. Firstly, we assign naive natural number indices to all patches making the sampled patch information devoid of any sense of location in the 2D image. Secondly, we assign the coordinates of the patch-centroid as patch-indices thus giving a sense of location only in a crude manner. In contrast to the baselines, our Hierarchical Patch-wise Spatial Sampling captures the best of both sparse and dense sampling strategies, indicating our method's superiority.
>
> **Q5. The authors propose a very complicated coordinate embedding scheme motivated by the ablation results in Table 8. But I wonder if there are not simpler methods with the same performance.**\
> **A5:** To the best of our knowledge, many of the existing works on INR literature have not explored the effect of introducing multiple types of coordinate embeddings and their fusion. We have designed these embeddings by keeping in mind the unique advantage of parameter efficiency along with the gains in performance. We believe that the intricate nature of our approach is justified by the observed performance gains, especially as demonstrated in the ablation experiments.
>
> **Q6. Formatting:**
> *  **Text in tables is way too small.**
> * **Weird parts of the text are in red. (around eq. 8, Table 6. and Table 9.)**
> * **PSNR is crossed in Table 9. What is this supposed to indicate?**
> * **Please use larger images in Figure 5. If in need for space, only show specific crops of the image.**
>
> **A6:**
> * To enhance the readability of the tables, we have increased the font size in table 1 by paraphrasing the longer definitions.
> * The text in red was a standard technique used in journals and conference papers for the revision cycles. However, to avoid any further issues, we convert the text in black.
> * Crossed out PSNR in Table 9 was a typo. We have fixed it.
> * We understand your concern regarding the size of the images in figure 5. We have moved this image to the appendix A5 so that it can be viewed at a larger scale.
>
> We are grateful for the time and energy the reviewer devoted in improving our submission. We have responded to all concerns and revised the manuscript accordingly. Nevertheless, if the reviewer has any remaining concerns, we are happy to address them.

---

> > ### Comment · Reviewer_LSEQ · 2024-01-09
> > **Thanks for addressing questions and concerns. There are still some formatting issues.**
> >
> > I would like to thank the authors for their elaborate response and the changes made to the paper. Most of my concerns seem to have been addressed.
> >
> > I still think there are some minor formatting issues.
> >   - The tiny fontsizes used in Table 1, 8 and 9 really reduce the readability of the paper.
> >
> >      *Suggestion: Use a single fontsize for tables throughout the paper. The fontsize used in Tables 2-5 is fine.*
> >   - Figure 6 is still inadequate. The task here is super resolution, so in order to judge the quality we need to show examples at a very high resolution. Instead a 1920×1080 video is displayed here as a 4x2cm image.
> >
> >      *Suggestion: Use full page width for this figure, or only display the crops in the main paper.*

---

> > > ### Author Response · Authors · 2024-01-13
> > > **Fixed the formatting issues**
> > >
> > > We appreciate the reviewer finding our responses meaningful. We have updated Tables 1, 8, 9, and Figure 6 for better readability in the revised version as suggested. We thank the reviewer for the thorough and insightful review.

---

> > > > ### Author Response · Authors · 2024-01-15
> > > > **Readability of Table 1**
> > > >
> > > > We have further enhanced the readability of Table 1 in the latest revision per the reviewer's kind suggestion.

---

### Review · Reviewer_xD4r · 2023-12-23

**Summary Of Contributions:**

1. This paper provides a novel solution to balance between inference efficiency and spatiotemporal continuity for video modelling with INRs by hierarchically sampling patch-wise spatial coordinates.

2. The work applies PNNs to process temporal signals and make adaptations to leverage PNNs’ strength in parameter efficiency.

**Audience:**

Yes

**Broader Impact Concerns:**

I have no concerns about the ethical implications of the work.

**Claims And Evidence:**

Yes

**Requested Changes:**

Minor issues:
- Two places refer to the same footnote 3 in Section 2. Please delete one of the references and leave no space between the text and the index.

- There’s no Section 8 but Section 3.1 and 3.2 refer to it.

- I find no description of NeRV**-L**.

- In Figure 6, ”Poly-INR” should be changed to “PNeRV” or “Ours” for consistency.

- Section 4.2.2: “As reported in Table 3, for SR, we compare our results with bicubic interpolation, INR-V (Sen et al., 2022), ZSSR (Assaf Shocher, 2018), and SIREN (Sitzmann et al., 2020b).” However, INR-V’s results are not presented in Table 3.

- Section 4.2.2: ”We also provide reasons for not comparing our results with VideoINR (Chen et al., 2022b), an important contemporary INR based method in Video SR in Appendix (Chen et al., 2022b).“ The referencing is incorrect.

**Strengths And Weaknesses:**

Strength

- The motivations of the proposed methods are presented clearly. I have little expertise in NeRV and PNNs, but the authors did a good job of helping me understand the context of this work and the reasons why they proposed these methods.

- The authors did extensive ablation studies to show the effectiveness of each proposed component.


W

- There are quite a few minor issues in the present paper (see Requested Changes below for several). I request the authors recheck the whole paper and try to fix the minor issues as much as possible, so the paper can be read more easily.

- The qualitative results are not very impressive to me. For example, the interpolation result in Figure 5 might be too sharp to be desirable.

---

> ### Author Response · Authors · 2024-01-06
> **Response to Reviewer xD4r**
>
> We are grateful to the reviewer for taking the time to provide important suggestions for improving our submission. We have incorporated the suggestions in the revised manuscript. Our responses to the reviewer’s specific concerns are as below:
>
> Q1. The qualitative results are not very impressive to me. For example, the interpolation result in Figure 5 might be too sharp to be desirable.\
> A1. We understand the importance of visually compelling results, and we acknowledge your observation that the interpolation result in Figure 5 might appear too sharp. It's crucial to emphasize that, as indicated in Table 4, the elevated PSNR values imply a superior overall quality of the interpolated video compared to existing methodologies. These high PSNR values signify a close resemblance between the interpolated video and the ground truth video. We also assess the quality of the interpolated video using multiscale structural similarity (MS-SSIM) metric. Unlike PSNR, MS-SSIM accounts for luminance, contrast and structure of each frame and hence better correlates with human perception. Our method outperforms existing methods on this metric as well as given below indicating that the interpolated video is closer to the original video.
>
> |   Method  |  MS-SSIM  |
> | -------   |  -----------  |
> |  NeRV-L   |    0.8161    |
> |  E-NeRV  |    0.9745     |
> |  Ours       |    0.9775     |
>
> To augment the above quantitative results, we present additional qualitative samples for illustrations [here](https://imgur.com/a/700UAv0).
>
> Q2. Two places refer to the same footnote 3 in Section 2. Please delete one of the references and leave no space between the text and the index.\
> A2: Thank you for bringing this to our attention. It has been rectified.
>
> Q3. There’s no Section 8 but Section 3.1 and 3.2 refer to it.\
> A3: Thanks for pointing this out. It's now accurately reflected as Table 8 instead of Section 8.
>
> Q4. I find no description of NeRV**-L**.\
> A4: The NeRV paper (cited below) proposes three variants namely NeRV-S, NeRV-M and NeRV-L  each with a varying number of learnable parameters. Among these variants, NeRV-L has the largest number of parameters i.e. 12.5 million parameters. To build the NeRV model with different sizes, the authors have changed the hidden dimension of the Multi layer perceptron and channel dimension of NeRV blocks. We have used NeRV-L for all our comparisons since our model’s size is comparable to that.
>
> NeRV:  “Hao Chen, Bo He, Hanyu Wang, Yixuan Ren, Ser-Nam Lim, and Abhinav Shrivastava. NeRV: Neural representations for videos. In A. Beygelzimer, Y. Dauphin, P. Liang, and J. Wortman Vaughan, Advances in Neural Information Processing Systems, 2021.”
>
> We have updated “NeRV” to “NeRV-L” at all instances of occurrence in the paper. Namely, in Table 2, Section 4.1, Figure 5, Table 4, Table 5, and Table 13.
>
> Q5. In Figure 6, ”Poly-INR” should be changed to “PNeRV” or “Ours” for consistency.\
> A5: Thank you for bringing it to our notice. We have addressed this concern by modifying “PolyINR” to “PNeRV”.
>
> Q6. Section 4.2.2: “As reported in Table 3, for super-resolution, we compare our results with bicubic interpolation, INR-V (Sen et al., 2022), ZSSR (Assaf Shocher, 2018), and SIREN (Sitzmann et al., 2020b).” However, INR-V’s results are not presented in Table 3.\
> A6. We are thankful to the reviewer for the attentive reading and pointing this out. We would like to clarify that there was an error in our previous statement regarding the comparison with INR-V (Sen et al., 2022). Contrary to what was initially mentioned, we do not compare our method with INR-V. We have corrected this now. As outlined in Appendix A.9, our principal reasons for not being able to draw comparisons between our method and INR-V for super-resolution are the following:
> * VideoINR uses ground truth High-Resolution (HR) video frames for training, while ours is a fully unsupervised approach utilizing only the low-resolution video for training.
> * Our method is a multifunctional INR. In that, it learns to represent a signal (video) as model weights. In contrast, VideoINR is an autoencoder trained specifically for Super-Resolution. Wherein, the claimed INR components function as non-linear transformations in the intermediate feature space.
> * INR-V's model, being a hypernetwork, faces limitations in handling high-resolution videos like the ones in the UVG dataset due to the unstable training routines associated with large hypernetworks.\
> Consequently, a comparison with INR-V becomes unfair.
>
> Q7. Section 4.2.2: ”We also provide reasons for not comparing our results with VideoINR (Chen et al., 2022b), an important contemporary INR based method in Video SR in Appendix (Chen et al., 2022b).“ The referencing is incorrect.\
> A7. We have fixed the referencing issue by appropriately referring to Appendix A.9.

---

### Decision · Action_Editor_Q6ix · 2024-01-23

**Recommendation:** Accept with minor revision

**Comment:**

All three reviewers recommended acceptance after the author responses. The reviewers do believe that presentation of the core ideas can be improved. So the decision is acceptance, but we strongly encourage the authors to not only incorporate the responses in the text, but also try to improve the clarity of the presentation of the method.

**Audience:**

Implicit Neural Representations are a powerful tool and this paper proposes a novel representation for videos. The part of the research community interested in INRs, video compression and beyond, would find this interesting.

**Claims And Evidence:**

Claims and evidence are both clear.

---

> ### Author Response · Authors · 2024-02-17
> **Camera Ready Revision**
>
> We thank the action editor and reviewers for helping us improve the paper throughout the feedback period. We believe that the feedback contributed to enhancing the clarity and soundness of the paper. We have updated the camera ready version. We have incorporated the responses to the reviewers in the text.